# Questioning the Endorheic Paradigm: Water Balance dynamics in the Salar del Huasco basin, Chile

Francisca Aguirre-Correa[1,2], Oscar Hartogensis[3], Pedro Bonacic-Vera[1], and Francisco Suárez[1,4,5]

[1]Department of Hydraulic and Environmental Engineering, Pontificia Universidad Católica de Chile, Santiago, Chile
[2]School of GeoSciences, University of Edinburgh, Edinburgh, UK
[3]Meteorology and Air Quality,Wageningen University, Wageningen, Netherlands
[4]Centro de Desarrollo Urbano Sustentable (CEDEUS), Santiago, Chile
[5]Centro UC Desierto de Atacama (CDA), Santiago, Chile

**Correspondence:** Francisca Aguirre-Correa (faguirre2@uc.cl)

**Abstract.** Arid endorheic basins exhibit limited water availability shaped by strong precipitation and evaporation variability. Understanding these processes is crucial for sustainable water resources management in such fragile environments. This study examines how rainfall and evaporation drive the spatial and temporal dynamics of groundwater recharge and water balance in an arid endorheic basin, using the Salar del Huasco in the Chilean Altiplano as a case study. For this, we implemented a

modified semi-distributed rainfall-runoff model integrated with a 40-year record (1980-2019) of satellite-derived precipitation and evaporation estimates. Results show that, on average over the catchment, about 12% of total rainfall (17 mm year$^{-1}$) recharges the aquifers, with a ∼35-day lag between rainfall and peak groundwater recharge. Spatial analysis reveals that most water infiltrates and recharges the groundwater system at high elevations (∼65% of total recharge), while low-lying wetlands, shallow lagoons, and riparian zones lose most of the water via evaporation (up to 950 mm year$^{-1}$). Our findings highlight that

when summer rainfall ceases, groundwater becomes the main water source supporting high evaporation rates, while recharge reaches a minimum by the end of autumn that persists until the end of the year. These results suggest a trade-off between groundwater recharge and evaporation for available water during the dry season. Moreover, while the basin receives around 145 mm year$^{-1}$ of annual precipitation, evaporation reaches 230 mm year$^{-1}$. These values imply a substantial water loss or an unaccounted groundwater inflow, challenging the endorheic assumption of the basin's hydrogeological boundaries. Future re-

search should revisit this assumption and incorporate fully coupled groundwater-surface water simulations to explicitly include interactions with lateral groundwater flows and groundwater levels, as well as with snow dynamics and vegetation processes currently omitted. Also, validating satellite-derived inputs against additional local observations is essential to strengthen the reliability of the water balance assessment. Nonetheless, these results provide a valuable framework and a first-approximation for quantifying water balance components in an arid basin, offering insights for water resource management in a context of

water scarcity and climate change.

# 1 Introduction

Arid regions are among the most hydrologically fragile environments on Earth, characterised by scarce water resources, elevated evaporation rates, and pronounced spatiotemporal variability in precipitation (Bouchaou et al., 2024; Scanlon et al., 2006). In these regions, the water balance is controlled by complex interactions between atmospheric, surface, and subsurface processes, making them particularly sensitive to climate fluctuations (Houston, 2006).

Within arid and semi-arid landscapes, a large fraction of drainage is classified as endorheic, i.e. internally drained basins where water is retained within topographic boundaries and lost primarily through evaporation (Gao et al., 2018). This endorheic paradigm has traditionally provided a simplifying framework: precipitation, surface runoff, and groundwater flows converge to a terminal lake, wetland, or salt flat, with no surface or subsurface outflows across the basin boundaries (Yapiyev et al., 2017). Consequently, water budgets in endorheic systems are generally assumed to be closed at the catchment scale, with evaporation and, in some cases, groundwater storage changes balancing the atmospheric inputs (Petch et al., 2023; Liu, 2022). However, increasing evidence suggests that endorheic basins are not always fully closed. Subsurface groundwater exchanges can connect apparently isolated basins, with implications for water availability, ecosystem persistence, and even transboundary water conflicts (Moran et al., 2019; Alcalá et al., 2018; Belcher et al., 2009). This calls for an important re-examination of the endorheic paradigm, especially in data-scarce arid regions where evaporation is intense and subsurface processes remain poorly constrained. The relevance of this issue extends beyond drylands, as endorheic basins collectively account for nearly 21% of the global continental land surface (Li et al., 2017).

The Altiplano, a high-altitude plateau in South America (Fig. 1a), is composed of multiple endorheic basins and is an example of an arid environment where water availability is both limited and largely dependent on intermittent, seasonal rainfall. Precipitation in this region occurs predominantly during the austral summer (December-February), driven by moisture transported westward from the Amazon basin (Garreaud et al., 2003). At a local scale, short-lived but intense convective storms are shaped by orographic effects, leading to a heterogeneous distribution of rainfall (Falvey and Garreaud, 2005; Vuille, 1999). However, most of this precipitation is ultimately returned to the atmosphere through evaporation processes, making it the principal mechanism of water loss in these basins (Johnson et al., 2010). Such losses occur across soil surfaces, open water bodies, and vegetated zones. However, wetlands, lagoons, and riparian zones account for the highest evaporative fluxes, yet persist year-round due to groundwater seepage and upwelling (Houston, 2006; Risacher et al., 2003).

While precipitation and evaporation represent the atmospheric components of the basin's water balance, groundwater recharge acts as a key redistributor of available water at the soil-atmosphere interface. Water from sporadic, intense high-altitude rainfall events infiltrates and moves through subsurface pathways, ultimately reaching lower-altitude zones where groundwater upwells and evaporates (Suárez et al., 2020). Characterizing the groundwater component of the water balance is therefore crucial for modelling and predicting the hydrological behaviour of arid systems. Despite its recognised significance, the spatiotemporal variability of groundwater recharge and the hydrological dynamic in groundwater upwelling areas remain poorly understood compared to the atmospheric components of the water balance. As a result, it is also necessary to revisit the endorheic assumption of these basins, which traditionally treats them as completely closed hydrological systems. Addressing this knowledge gap

is increasingly urgent in light of projected climatic changes and intensifying anthropogenic pressures, both of which stand to intensify water scarcity across the Altiplano (Satgé et al., 2019).

In this study, we use the Salar del Huasco basin in the Chilean Altiplano (Fig. 1b) to investigate how precipitation and evaporation interact to control groundwater recharge and upwelling, within the traditionally assumed endorheic paradigm. In particular, we aim to address the following question: *To what extent do spatial and temporal variations in precipitation and*
*evaporation regulate groundwater recharge in an arid endorheic basin, and do observed imbalances challenge the assumption of hydrological closure?*

To address this research question, we use a rainfall-runoff model built on the work of Uribe et al. (2015). This model was developed for the Salar del Huasco and simulates the basin's continuous hydrological response using daily precipitation, mean air temperature, and potential evaporation records, distributed spatially over the basin according to topographic gradients. Here,
we adapt the rainfall-runoff model to estimate groundwater recharge based on satellite-derived input data, thereby capturing the spatial heterogeneity in the atmospheric forcings with better resemblance to reality than using topographic gradients. We also further modify the model to study groundwater evaporation - recharge trade-off for moisture and to identify evaporation sources through a simplified back-trajectory analysis.

The remainder of this manuscript is structured as follows: Sect. 2 describes the study area and methodological framework;
Sect. 3 presents and discusses the main findings. Sect. 3.1 analyses the estimated groundwater recharge results, Sect. 3.2 explores the role of groundwater in sustaining evaporation, with special focus on areas of groundwater upwelling, and Sect. 3.3 provides a general discussion on the basin's overall water balance; finally, Sect. 4 summarises the key findings and their implications for hydrological modelling and water resources management in arid regions.

## 2 Methods

### 2.1 Study site


The Salar del Huasco is an arid, endorheic basin located between 19°54' - 20°27' S and 68°40'- 69°00' W in the Chilean Altiplano (Figs. 1a,b). Covering an area of approximately 1470 km$^2$, the basin lies at an average elevation of 4165 m above sea level (asl), and forms part of the Depresión de los Salares, a geomorphological depression surrounded by mountain ranges with elevations between 4000 and 5200 m asl. At the basin's lowest topographic point, there is a salt flat nucleus that extends
over approximately 50 - 60 km$^2$ comprised by wetlands and a shallow, salty lagoon (Fig. 1c). The shallow lagoon experiences important seasonal fluctuations in its extension (Lobos-Roco et al., 2021). The main surface water flow in the basin is the Collacagua River, which runs north to south for roughly 15 km. About 10 km upstream of the Salar del Huasco salt flat, the river infiltrates into alluvial deposits, leaving the final stretch of its course subsurface under normal flow conditions. However, during intense rainfall events surface runoff may extend to the salt flat (Uribe et al., 2015).
Around the margins of the salt flat nucleus, groundwater discharges primarily via six main springs (Fig. 1b). Long-term gauging station data from the Chilean National Water Division (Dirección General de Aguas, DGA) demonstrate relatively stable spring flow year-round, with annual average discharges between $8 \times 10^{-3}$ m$^3$s$^{-1}$ and $24 \times 10^{-3}$ m$^3$s$^{-1}$ (DGA, 2009).

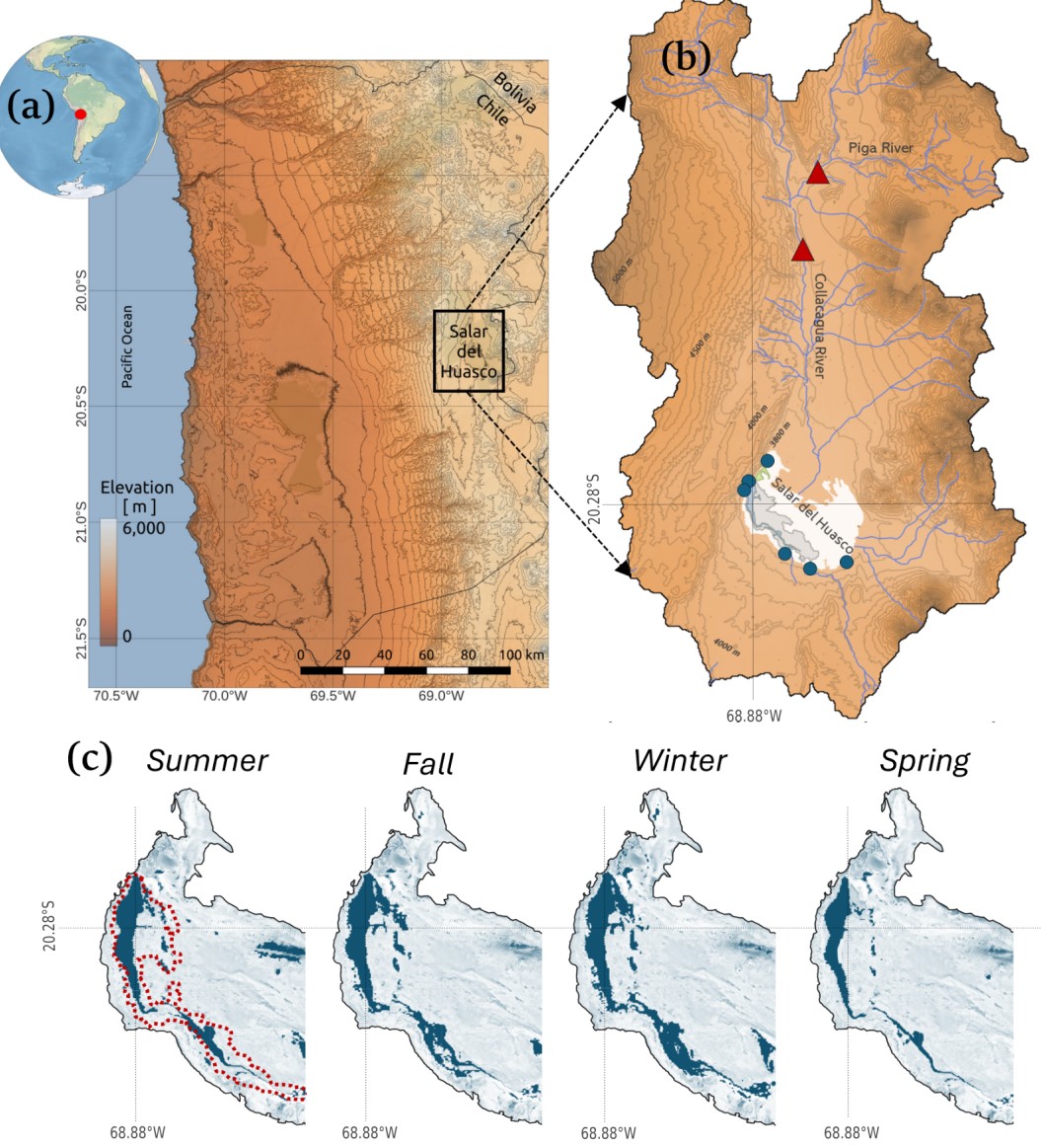

**Figure 1.** (a) Geographic localization of the study area, the Salar del Huasco basin, in the Chilean Altiplano. Line contours indicate altitude levels every 250 m. (b) Zoomed-in view of the Salar del Huasco basin with the location of fluviometric stations in the Collacagua and Piga River (red triangles) and groundwater-fed springs (blue circles) that discharge to the salt flat nucleus at the basin's lowest point. Line contours are altitude levels every 50 m. (c) Seasonal fluctuations of the shallow, salty lagoon, located in the main wetland, during 2018. Red dotted lines in summer indicate lagoon extent in 2019, a year with above-average precipitation. Lagoon area was estimated based on Modified Normalized Difference Water Index (MNDWI) from Landsat data.

In total, they contribute to the salt flat water balance with $\sim 0.2$ m$^3$s$^{-1}$, equivalent to an order of magnitude of $\sim 10^2$ mm year$^{-1}$. Spring flow has been recognised as a key contributor to vegetation growth and the formation of the shallow, salty lagoon, from which water continuously evaporates (Lobos-Roco et al., 2022; de la Fuente and Meruane, 2017).

## 2.2 Modified rainfall-runoff model

To estimate groundwater recharge in the Salar del Huasco basin, we used a modified version of the Uribe et al. (2015) semi-distributed rainfall-runoff model, which was developed based on the soil-moisture accounting routine of HEC-HMS (Feldman, 2000; Hydrologic Engineering Center US, 2001). The rainfall-runoff model estimates key hydrological processes at a daily timescale by partitioning precipitation among different conceptual reservoirs using a set of simplified water balance equations, including surface runoff, infiltration, evaporation, and groundwater flows (Fig. 2a). We define groundwater recharge as the subsurface flow reaching the groundwater reservoir, which can become deep percolation or baseflow (Blin et al., 2022).

The spatial domain of the model is organised into 21 sub-basins, each divided into one or more Hydrological Response Units (HRUs) with homogeneous geological features (Leavesley et al., 1983) (Fig. 2b). Five HRUs were defined, each characterised by distinct hydrogeological properties and spatial connectivity derived from a 90 m-resolution digital elevation model (Shuttle Radar Topography Mission, SRTM). HRU R1 is defined as rocks with very low effective porosity and without fractures (57% of the total area), R2 represents fractured rocks with significant effective porosity (14% of the total area), S1 represents lacustrine and evaporite deposits (17% of the total area), S2 corresponds to alluvial and colluvial deposits (7% of the total area), and S3 represents fluvial deposits (5% of the total area) (Uribe et al., 2015).

A total of 15 parameters govern the water balance computations in each HRU, which are calibrated using streamflow records from Piga and Collacagua River DGA stations (Fig. 1b). Thus, the model provides inferred recharge fluxes based on surface-water calibration (Appendix A). While the lack of reliable groundwater data precludes explicit calibration of groundwater recharge, we note that this model has been previously applied in the Salar del Huasco and other Altiplano basins (e.g., Yáñez-Morroni et al. (2024a, b); Blin et al. (2022); Acosta (2004)). In these studies, recharge outputs from the rainfall-runoff model were successfully incorporated into groundwater flow models, demonstrating its credibility in groundwater applications. For instance, the Uribe et al. (2015) model was applied by Blin et al. (2022) to estimate aquifer recharge in the Salar de Huasco, providing a spatiotemporal distribution of recharge consistent with groundwater-well observations. Here, we primarily use it to represent the timing and partitioning of surface runoff and recharge.

For this study, we adapted the model in two ways to improve the spatiotemporal characterization of groundwater recharge. Modifications to the model are detailed in the following subsections. To evaluate the basin's daily to interannual hydrological response to precipitation and evaporation variability, we applied the modified model over a 40-year period (1980 - 2019). The model calibration results are presented in Appendix A. A full description of the model's governing equations can be found in Blin et al. (2022).

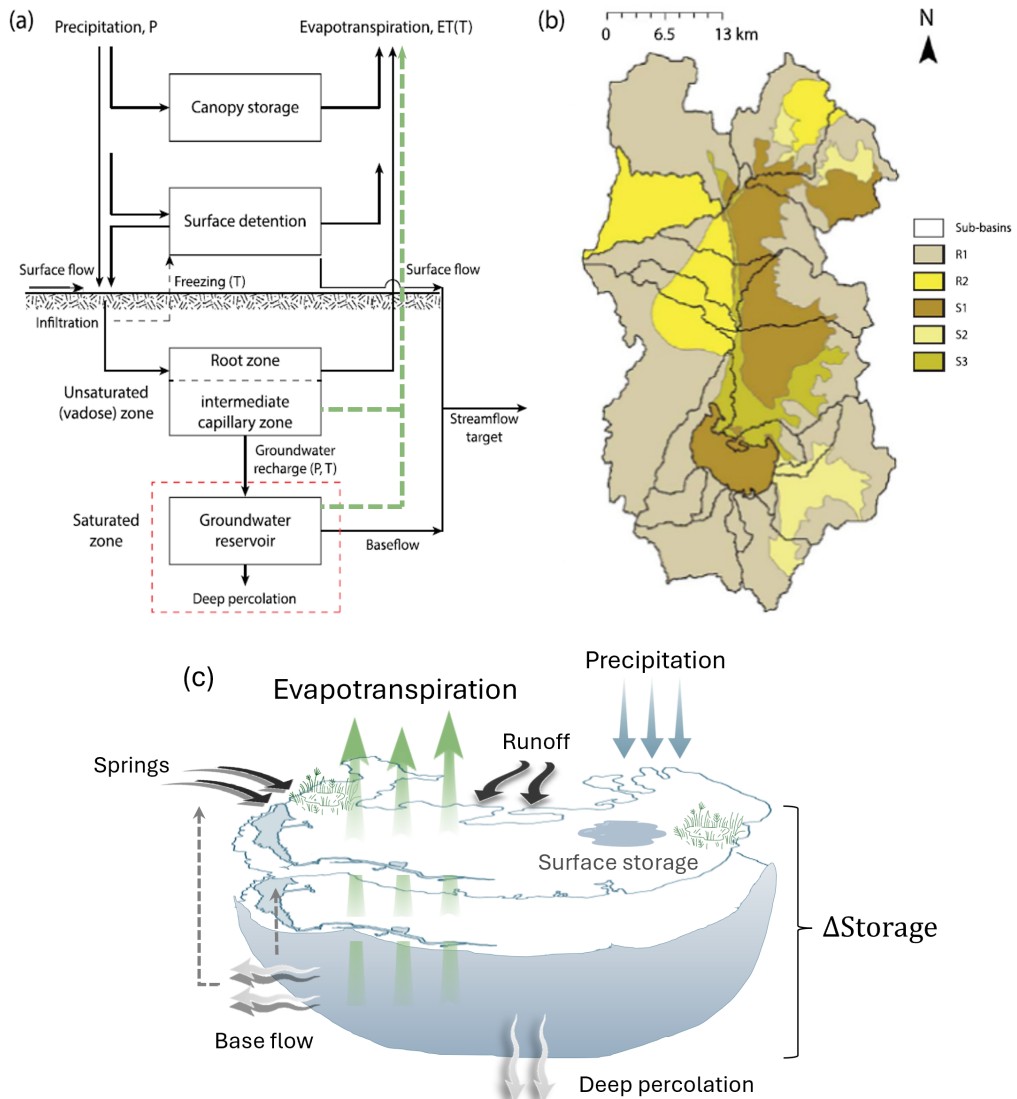

**Figure 2.** (a) A schematic representation of the rainfall-runoff model reservoirs and the fluxes involved within each Hydrologic Response Unit (HRU) in the Salar del Huasco basin. For this study, we introduce explicit evaporation from the unsaturated zone and groundwater reservoirs (green arrows). (b) Model spatial domain composed of five HRUs and 21 sub-basins that contribute to the primary drainage network of the basin. (c) Schematic of water balance in the salt flat nucleus sub-basin, in which external springs flow, precipitation and runoff from neighbouring sub-basins represent the inflows to the salt flat nucleus, while evaporation and deep percolation represent the outflows. Since the salt flat nucleus is the lowest-point sub-basin, generated runoff represents accumulation in ponds, the lagoon or wetlands as surface storage, while base flow is recirculated groundwater that can either contribute to seepage or spring discharge. Panels (a) and (b) were modified from Blin et al. (2022).

### 2.2.1 Satellite-derived datasets for input variables

We first modified the Uribe et al. (2015) model by integrating daily precipitation, air temperature and evaporation gridded data instead of using the original topographic-gradients approach. This improvement allows for a better representation of the hydrometeorological input data, and a more realistic characterization of groundwater recharge variability in space and time.

For precipitation and air temperature, daily data at ∼ 5 km spatial resolution from the Center for Climate and Resilience Research Meteorological (CR2MET) data version 2.0 are used (Boisier et al., 2018). For evaporation, we implement the

Earth Engine evaporation Flux (EEFLUX) tool, which applies the Mapping evaporation at high Resolution with Internalized Calibration (METRIC) model to Landsat images (30 m resolution every ∼16 days) to estimate evaporation (Allen et al., 2007). In this study, evaporation refers specifically to satellite-retrieved actual evaporation (ETa) from EEFLUX/METRIC, rather than potential evaporation or other model-estimated fluxes.

Since the rainfall-runoff model results depend directly on the quality of the forcing data, we applied bias corrections using

local ground-based observations to improve the reliability of the satellite products (Appendix B). A machine learning-based bias correction was developed to downscale CR2MET precipitation and temperature fields to the basin scale through bias correction, using ground-based observations from DGA meteorological stations distributed across the Salar del Huasco basin (see Figs. B1 and B2). EEFLUX evaporation estimates were temporally disaggregated to daily resolution and further corrected through a machine learning approach constrained by local meteorological inputs (Figs. B3 and B4). For details regarding the

datasets, including their description, processing and validation for the Salar del Huasco basin, we refer to Appendix B.

### 2.2.2 Integration of groundwater evaporation

In arid environments, residual moisture often persists in the unsaturated zone after rainfall ceases and continues to evaporate in the absence of direct precipitation (Sutanudjaja et al., 2014). Similarly, in areas where the water table is shallow (e.g., lagoons, wetlands, riparian zones), groundwater can supply moisture to the unsaturated zone via capillary rise, as long as a hydraulic

connection exists within the soil pores, or evaporate directly when the saturated zone is exposed to the atmosphere (Hernández-López et al., 2014; de la Fuente et al., 2021). Although water evaporates at the surface, these evaporation processes imply water losses from their respective reservoirs, altering soil moisture profiles and modulating groundwater recharge. This is particularly relevant in the Salar del Huasco basin, where the highest evaporative demand occurs approximately eight months after the rainy season (Lobos-Roco et al., 2022), with groundwater serving as the primary moisture source for evaporation.

The Uribe et al. (2015) rainfall-runoff model is driven solely by precipitation, meaning that evaporation occurs only in response to rainfall events. To better represent the study site evaporation dynamics, we adapted the model by introducing an explicit representation of evaporation from the unsaturated zone and shallow groundwater reservoirs (green arrows in Fig. 2a). With this we aim to (1) more accurately capture the interactions between groundwater recharge and evaporation fluxes, (2) improve estimates of groundwater recharge by accounting for the trade-off between recharge and evaporation for available

moisture, and (3) identify evaporation sources through a simplified back-trajectory analysis.

Our modified model uses the remotely sensed observations of actual evaporation to estimate unsaturated zone and groundwater evaporation as residual processes. Limited by each reservoir's available water, this is the remaining observed evaporation after removing evaporative losses from previous reservoirs. This approach is already implemented in the model to estimate root zone evaporation (see Fig. 2a), so it is only extended to deeper reservoirs. In the model, capillary rise is not simulated explicitly, so evaporation from the unsaturated zone is simplified to depend only on moisture availability following rainfall events. Groundwater recharge mainly occurs at high elevations, where capillary rise is negligible due to deep water tables. In shallow water tables, capillary water is likely to be rapidly lost to the atmosphere given the high evaporative demand, rather than remaining in the soil for extended periods. This simplification therefore does not introduce large errors in our recharge estimates. Note that, under this definition, evaporation from the unsaturated zone can occur in areas with deep water tables, as it represents moisture loss from residual soil water after rainfall events, independent of groundwater processes. Groundwater evaporation, instead, is introduced only where the water table is shallow enough to evaporate. For this, we use the region defined by Blin and Suárez (2023), which encompasses the salt flat nucleus and riverbeds. We refer to it as a "potential groundwater evaporation," as the model does not account for lateral groundwater inflow between sub-basins, groundwater level dynamics, or extinction depths, all of which are required to estimate actual groundwater loss (Shah et al., 2007).

Despite these simplifications, this approach is a first-order approximation to understanding interactions between groundwater evaporation and groundwater recharge, as well as to identifying the primary sources of evaporative water loss throughout the year. With this, we can evaluate the potential role of groundwater recharge and evaporation in controlling the basin's water balance.

## 2.3 Spatiotemporal analysis of Groundwater Recharge

Groundwater recharge in the Salar del Huasco is estimated using the Uribe et al. (2015) modified rainfall-runoff model results. For the temporal analysis, we analyse the basin hydrological response across different timescales, which corresponds to the sum of the individual HRU responses in each sub-basin (Legesse et al., 2003). We provide an overview of groundwater recharge variability from monthly to interannual timescales. Given the short-lived, yet intense convective storms that characterise the rainy season, we complement this analysis by performing a lagged correlation analysis between recharge and its atmospheric drivers at a daily timescale. With this, we intend to determine how recharge responds to the summer wet-season pulses and whether delayed groundwater recharge can persist during the drier months.

Given the heterogeneity in the sub-basins geology (Fig. 2b), we assess the spatial variability of groundwater recharge by analysing the modified model results in each HRU. In particular, we evaluate seasonal spatial fluctuations of groundwater recharge and its atmospheric drivers to evaluate how localised areas contribute to the annual cycle of the basin's hydrological response. With this, we also aim to determine whether recharge patterns respond primarily to patterns in precipitation, evaporation, or a combination of both.

## 2.4 Water balance analysis in Groundwater Upwelling zones

As the salt flat nucleus has been recognised as a highly evaporative ecosystem sustained by groundwater upwelling and spring discharge, we conducted a simplified water balance analysis in the salt flat nucleus sub-basin (Fig. 2c) to better understand the role of groundwater flows in driving its hydrological behaviour. Equation 1 defines its water balance:

$$\frac{dS}{dt} = P + Q_{runoff} + Q_{springs} - E - DP + SS + BF \tag{1}$$

where $\frac{dS}{dt}$ represents temporal changes in water storage, $P$ refers to precipitation, $Q_{runoff}$ to runoff, $Q_{springs}$ to external springs flow (i.e., originating outside the sub-basin), $E$ to total evaporation from the system, as observed by EEFLUX, and $DP$ to deep percolation. Since the salt flat nucleus is the lowest-point sub-basin, generated runoff in the model represents accumulation in ponds, the lagoon or wetlands as surface storage ($SS$), while base flow is recirculated groundwater that can either contribute to seepage or spring discharge ($BF$) (see Fig. 2c). All terms, except external spring discharge (i.e., inflow originating outside the analysed sub-basin), are inputs or outputs from the modified model. For external springs flow, we assume a constant value of 0.2 m$^3$s$^{-1}$ based on the hydrogeological reports from DGA.

In our analysis, the rainfall-runoff model treats the salt flat nucleus sub-basin as a single averaged domain when in reality it is comprised by wetlands, a saline, shallow lagoon, and salt surfaces (Suárez et al., 2020). To link changes in the salt flat nucleus sub-basin storage to actual fluctuations in the shallow lagoon, we provide an assessment of the primary hydrological fluxes that govern the lagoon dynamics. For this, we performed an orthogonal regression analysis between the lagoon area and the salt flat nucleus sub-basin's inflows and outflows (Fig. 2c). The lagoon area was estimated from daily interpolated Landsat images (see Appendix B for further details). This approach allows to assess whether daily area fluctuations are predominantly surface-driven (e.g., immediate responses to rainfall, runoff or surface evaporation) or if groundwater processes (e.g., springs flow, groundwater recharge or groundwater evaporation) play a more significant role in maintaining the shallow, salty lagoon.

Finally, we extend the water balance analysis to the basin scale, offering a comprehensive overview of the Salar del Huasco basin's key hydrological dynamics and evaluating the contribution of each component to the overall water budget.

## 3 Results and Discussion

### 3.1 Groundwater Recharge driven by Rainfall and Evaporation variability

Figure 3 presents the hydrological response of the Salar del Huasco basin according to the Uribe et al. (2015) modified model results. Groundwater recharge is displayed as monthly averages scaled to their annual equivalents in mm year$^{-1}$, along with precipitation and evaporation observations from the satellite-derived datasets.

The long-term annual mean groundwater recharge is estimated as 17 mm year$^{-1}$ (12% of the annual rainfall). At a daily timescale, precipitation events do not lead to an immediate groundwater recharge response (Fig. 3), suggesting a temporal lag between rainfall and recharge processes. This lag is driven by high evaporation rates occurring simultaneously with precipita-

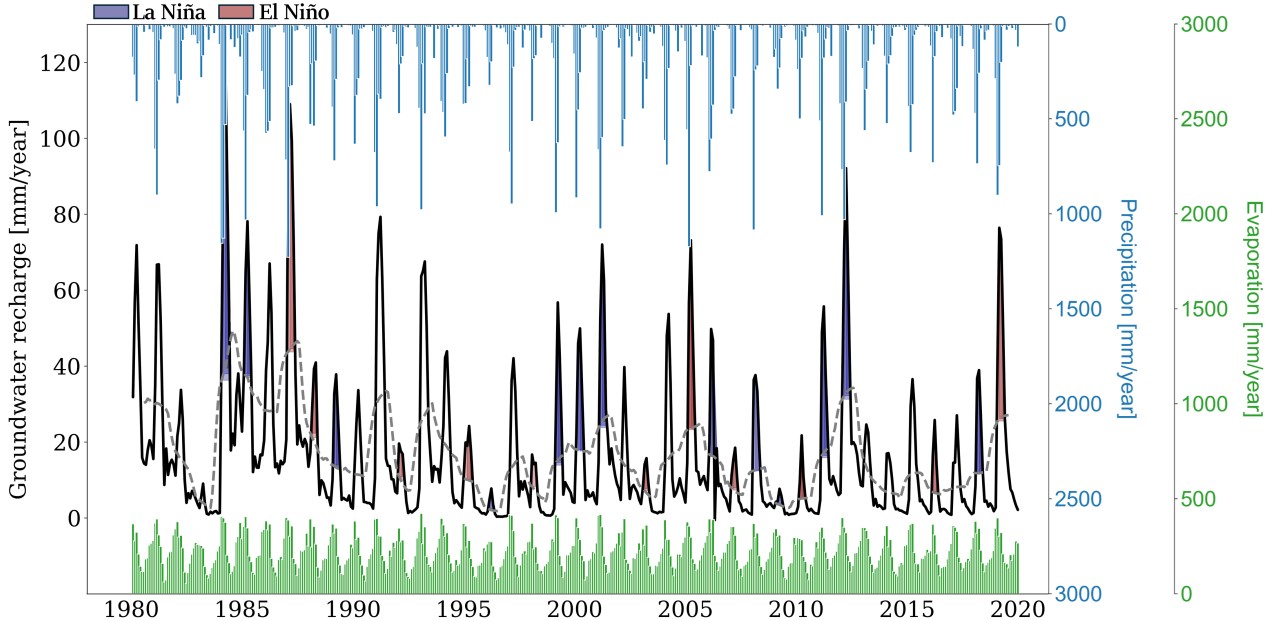

**Figure 3.** Groundwater recharge estimated for the Salar del Huasco basin using the Uribe et al. (2015) modified rainfall-runoff model. Values are presented for the 1980 - 2019 period and represent monthly means, scaled to annual equivalents (mm year$^{-1}$) for consistency across all water balance components. Groundwater recharge 12-month running average is displayed in grey dashed line. Blue shades correspond to La Niña years, while red shades correspond to El Niño years. Monthly averaged precipitation (blue bars) and evaporation (green bars) in mm year$^{-1}$ from gridded CR2MET and EEFLUX datasets, respectively, are also presented.

tion events (Fig. 3), and by a buffering effect in the vadose zone that delays percolation into the groundwater system, thereby modulating the net effective recharge. A daily lagged correlation analysis reveals a ∼35-day delay between net precipitation (rainfall minus surface evaporation) and groundwater recharge. Surface evaporation refers to the direct loss of water to the atmosphere from water stored at the land surface, including temporary surface ponding. This delay indicates that infiltration from rain does not immediately translate into maximum percolation to the groundwater reservoir. Instead, there is a mild correlation during the precipitation event ($r = 0.52$), which gradually intensifies over time until one month later net precipitation and groundwater recharge reach their strongest correlation ($r = 0.84$). From there on, the correlation decreases, but only changes its sign after ∼120 days. These results suggest a gradual distribution of percolation in which subsurface processes continue to recharge the groundwater reservoir for up to four months after a rainfall event occurs.

Seasonally, groundwater recharge rates increase during December-March, which corresponds to the summer rainy season, whereas lower recharge rates are observed from winter onward (Fig. 3). Beyond the seasonal cycle, groundwater recharge presents an important interannual variability, with alternating periods of high and low values that closely follow precipitation (Fig. 3). The major source of interannual variability in the Altiplano is related to the El Niño-Southern Oscillation (ENSO) (Garreaud et al., 2003; Garreaud and Aceituno, 2001). Results suggest that during La Niña years (blue shaded years in Fig. 3),

characterised by anomalous high precipitation in Salar del Huasco (Lobos-Roco et al., 2022), generally coincide with peaks in groundwater recharge (e.g., strong La Niña during summer seasons of 1984, 1985, 2001 and 2012). Conversely, during El Niño years there are drier conditions, which coincides with lower recharge values (e.g., strong El Niño during summer seasons of 1983, 1998 and 2010 shaded in red in Fig. 3). Regarding interdecadal variability, previous studies report recurrent wet years

every 20 - 30 years, enabling recharge of groundwater reservoirs that feed terminal wetlands within the Altiplano (de la Fuente et al., 2021). This dynamic is therefore crucial for the basin's water balance, as these occasional extreme wet years significantly recharge the aquifers, and thereby support evaporation and sustain ecosystems during extended dry periods. Even though we cannot evaluate interdecadal fluctuations in this study as longer datasets are needed, it is worth mentioning that the salt flat nucleus sub-basin has an annual mean recharge of $\sim 4$ mm year$^{-1}$, and only experiences significant recharge in those years in

which the Salar del Huasco basin hydrological response is larger than 70 mm year$^{-1}$. In particular, exceptional groundwater recharge ($\geq 80$ mm year$^{-1}$) is observed during the summer seasons of 1984 and 2012 in the salt flat nucleus sub-basin, which might suggest a recharge recurrence every $\sim 25 - 30$ years in line with previous studies. Further analysis is required to quantify these hydroclimatic teleconnections, but the observed patterns suggest a significant influence of climate-driven precipitation variability on long-term groundwater recharge and water balance dynamics in the Salar del Huasco basin.

To identify which sub-basins contribute the most to the Salar del Huasco basin hydrological response, and evaluate the role of precipitation and evaporation driving it, Fig. 4 presents the spatial seasonal means of precipitation, evaporation and groundwater recharge across the basin, along with the Salar del Huasco basin annual cycles. Values are scaled to their annual equivalents in mm year$^{-1}$.

  Rain is mainly concentrated in the summer season (DJF), with an annual mean of $\sim 145$ mm year$^{-1}$, and it is predominantly

observed in the eastern sector of the basin, particularly over high-altitude HRUs (Fig. 4a, see also Fig. 1b). Sub-basins above 4500 m asl have an average precipitation of $\sim 192$ mm year$^{-1}$ compared to a value of $\sim 127$ mm year$^{-1}$ in those sub-basins below 4000 m asl. This spatial pattern underscores the relevance of orography in shaping the endorheic basins convective storms. Moreover, its eastern-concentrated pattern aligns with the dominant moisture transport mechanism in the region, as rainfall in the Altiplano is modulated by eastward upper-level winds that advect moisture from the Amazon Basin (Garreaud

et al., 2003). As a result, precipitation decreases significantly from east to west (Vuille and Keimig, 2004). Outside the summer months, precipitation is nearly absent, with the exception of occasional winter precipitation events, as indicated by the shaded standard deviation in the annual cycle (right panel in Fig. 4a).

  The spatial distribution of evaporation (annual mean of $\sim 230$ mm year$^{-1}$) reveals that evaporative losses are most pronounced in low-elevation areas, particularly in the salt flat nucleus and in the streambed of the Collacagua river (Fig. 4b).

These sub-basins, characterised by shallow water tables, wetlands and lagoon, sustain high evaporation rates throughout the year (S1-type $\sim 550$ mm year$^{-1}$, S3-type $\sim 500$ mm year$^{-1}$). Among these, the salt flat nucleus sub-basin is particularly relevant, as it evaporates an annual mean of $\sim 950$ mm year$^{-1}$. Nevertheless, some high-altitude sub-basins in the northeastern sector of the basin also exhibit notable evaporation rates, despite their elevation ($> 4000$ m asl). This can be attributed to two main factors: (1) the presence of lacustrine and evaporite deposits (classified as S1-type HRUs with an annual mean of $\sim 600$

mm year$^{-1}$, see Fig. 4b) and (2) intense precipitation events ($\sim 200$ mm year$^{-1}$) in these sub-basins during summer (Fig.

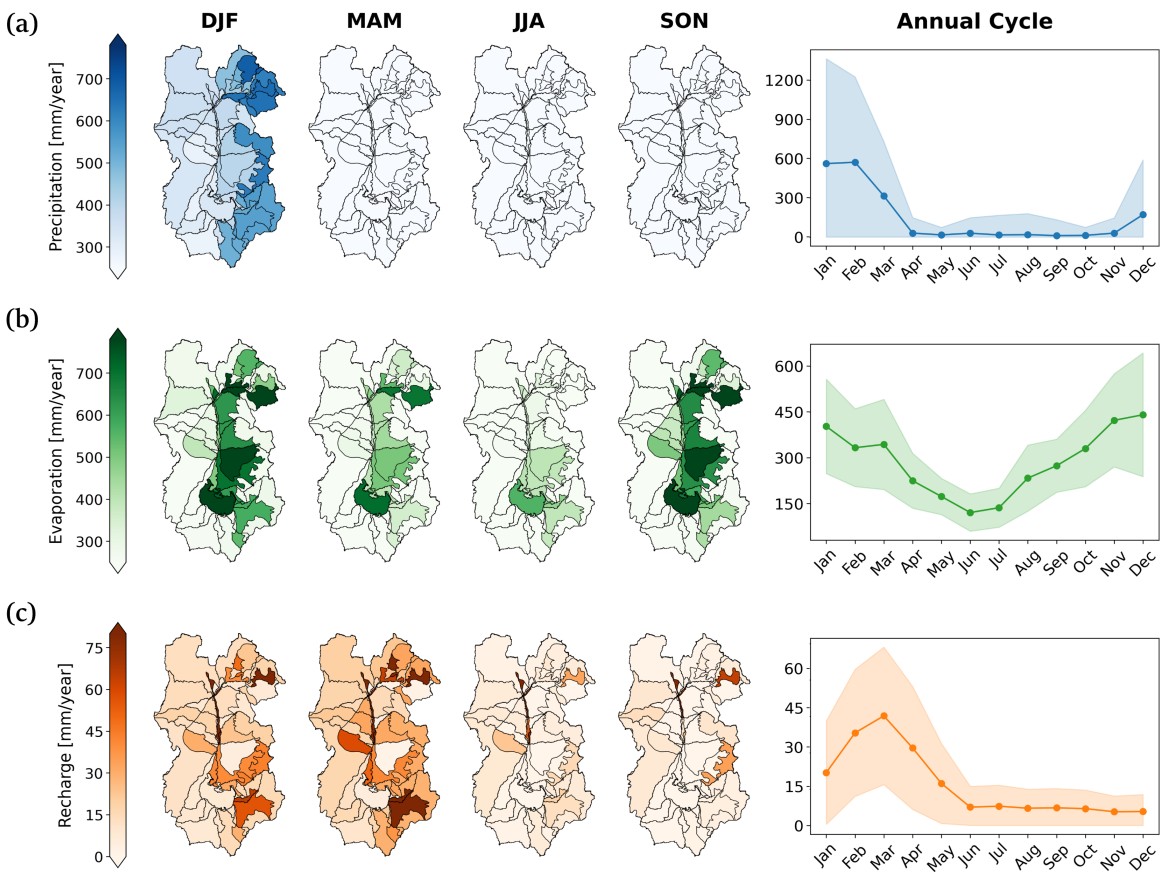

**Figure 4.** Left panels: Spatial distribution of seasonal means across sub-basins in mm year$^{-1}$ for (from left to right) DJF, MAM, JJA and SON for (a) Precipitation, (b) Evaporation and (c) Groundwater recharge. Right panels: Monthly mean variability of each variable. Values represent monthly means, scaled to annual equivalents (mm year$^{-1}$) for consistency across all water balance components. Shades indicate the standard deviation. Rain and Total evaporation are observations from gridded CR2MET and EEFLUX datasets, respectively.

4a), ensuring water availability for evaporation. While maintaining the same spatial pattern, evaporation annual cycle (right panel in Fig. 4b) exhibits a pronounced seasonal variability, with evaporation rates responding to the interplay between water availability and solar radiation (Lobos-Roco et al., 2022). High evaporation rates are observed during summer (DJF) mainly due to water availability from rainfall events. However, the highest rates are observed during spring (SON), suggesting that evaporation is mainly driven not by direct water inputs from precipitation, but by an increase in atmospheric evaporative demand due to ideal conditions of vapor pressure deficit, thermal conditions for circulation patterns and radiation conditions due to the lack of cloud cover (Lobos-Roco et al., 2022). During the winter months (JJA), evaporation rates decline to their lowest levels, which can be attributed to reduced water availability following the summer season, and low temperatures associated to a weaker radiative forcing. Here, the salt flat nucleus is the main sub-basin sustaining the low, but existant, evaporation demand.


The spatial distribution of groundwater recharge does not exclusively follow precipitation patterns neither the evaporation ones (Fig. 4c). Groundwater recharge spatial variability is controlled by the interaction between both atmospheric components of the water balance, and further modulated by the geology and soil properties. The highest specific recharge rates, understood as the highest groundwater recharge per unit area, are found in the fluvial deposits formed around the northern section of the Collacagua River (HRU-type S3, $\sim$60 mm year$^{-1}$, Fig. 2b), which have high permeability and receive runoff contributions

from the surrounding sub-basins. The lowest specific recharge rates are found in the lower elevation parts of the basin, which are mainly HRU-type S1 ($\sim$7 mm year$^{-1}$). Besides not receiving much water from precipitation (Fig. 4a), they contribute little to recharge mainly due to their fine-grained texture and evaporitic nature, which limit infiltration and promote water retention near the surface. For instance, S1-type HRU located in the northeastern part of the basin does not contribute to recharge despite experiencing intense rainfall rates (Fig. 4a), as most of the water is returned back to the atmosphere through evaporation (Fig.

4b). In terms of volumetric flows, however, recharge mostly occurs in high-altitude sub-basins ($\sim$0.5 m$^3$s$^{-1}$, $\sim$15 mm year$^{-1}$), where convective storms are more intense ($\sim$200 mm year$^{-1}$, northeastern part of the basin). Yet, recharge primarily occurs on those places where evaporative losses remain relatively low. Therefore, despite being impermeable rocks, HRU-type R1 (Fig. 2b) contributes the most to the total groundwater recharge ($\sim$0.35 m$^3$s$^{-1}$, $\sim$12 mm year$^{-1}$) due to its location in high-elevation areas with larger rainfall rates, its low evaporative capacity, and also because it covers the largest area in the basin ($\sim$57%).

Note that in these high-altitude sub-basins no capillary rise exist due to deep water tables, which reduces the uncertainty of our recharge estimates as explained in Sect. 2.

    Controlled by this spatial pattern, the annual cycle of groundwater recharge (right panel in Fig. 4c) reveals a distinct seasonality, with a pronounced peak at March right after the rainy season. This corresponds to a transition period in which few rainfall events occur but lower evaporation rates are observed, thereby allowing water to infiltrate and recharge the aquifers. Following

this peak, groundwater recharge gradually decreases through autumn (MAM) until reaching minimum levels at winter (JJA), which persist in that condition until the end of the year due to high evaporation rates. This gradual decay after the rainy season suggests that water stored in the unsaturated zone continues to percolate downward, sustaining groundwater recharge even in the absence of direct precipitation inputs. These findings are consistent with the lagged correlation analysis previously presented, in which rainfall events can sustain groundwater recharge up to $\sim$4 months after a rainfall event, and highlights the

buffering capacity of the vadose zone in delaying precipitation impact over recharge.

    Despite these findings, it is important to acknowledge that our groundwater recharge estimates are subject to uncertainties related to both model structure and input data. The modified rainfall–runoff model offers a first-order approximation and relies on satellite-derived inputs that, despite local bias correction, retain spatial and temporal uncertainties, particularly in poorly monitored areas. These residual errors can propagate through the model and affect recharge estimates. In addition, dry-season

evaporation is prescribed using satellite estimates rather than dynamically simulated, which may misrepresent seasonal water losses and bias recharge estimates under dry conditions. The model also excludes snow accumulation, melt, and freeze-thaw dynamics, which, while not dominant, may influence infiltration timing in high-altitude areas. Furthermore, the 15-parameter calibration increases the risk of equifinality, potentially limiting the uniqueness of the recharge estimates. Lastly, recharge fluxes should be interpreted as model-inferred outputs constrained by surface flows rather than direct simulations of groundwater

levels. Despite these limitations, the model enables spatially distributed and process-oriented analysis of groundwater recharge, and provides a valuable baseline for understanding the hydrological dynamics in the Salar del Huasco basin under the current climatic and data constraints.

## 3.2 Groundwater sustaining Evaporation

To identify potential water sources that supply the Salar del Huasco basin high atmospheric evaporative demand, Fig. 5 differentiates evaporation losses from the surface, unsaturated zone and groundwater reservoir.

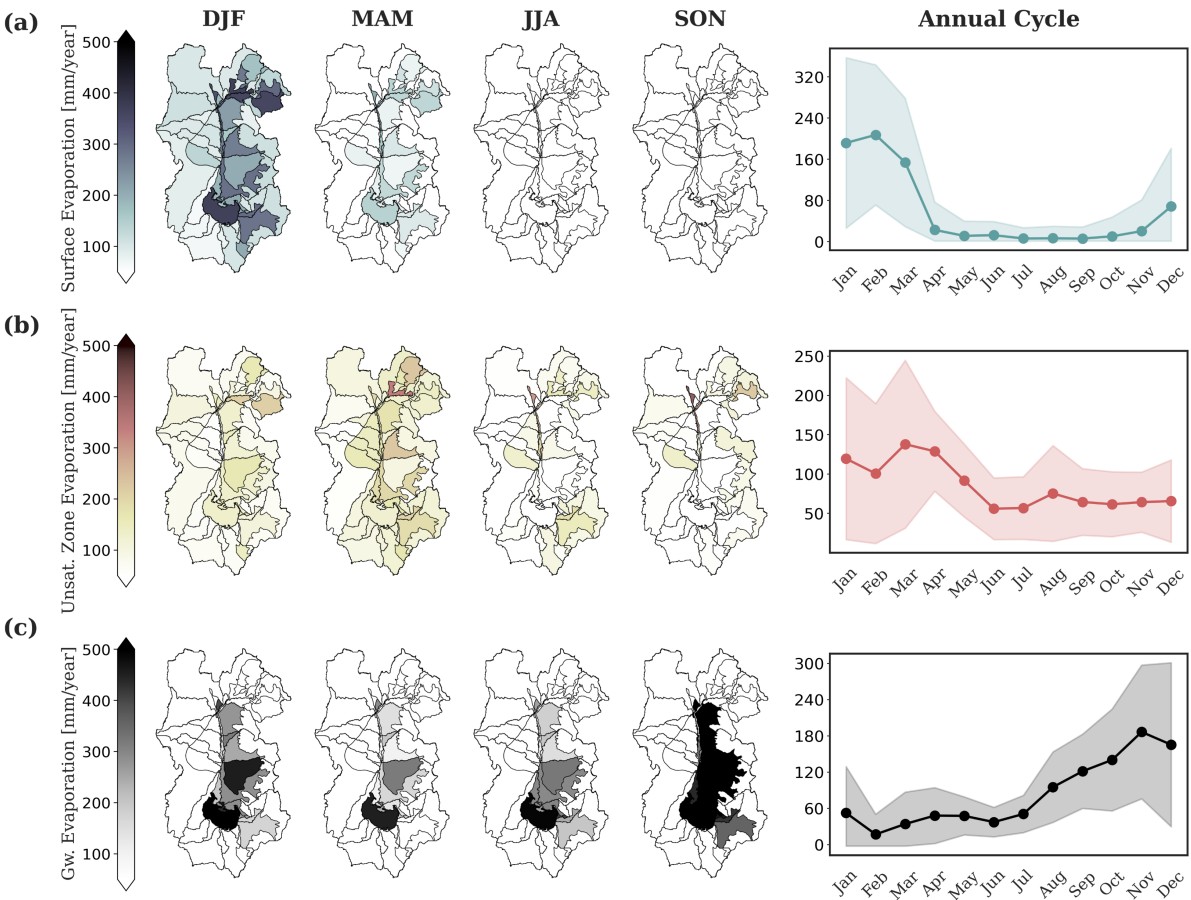

**Figure 5.** Left panels: Spatial distribution of seasonal means across sub-basins in mm year$^{-1}$ for (from left to right) DJF, MAM, JJA and SON for (a) Surface evaporation, (b) Unsaturated zone evaporation and (c) Groundwater evaporation. Right panels: Monthly mean variability of each variable. Values represent monthly means, scaled to annual equivalents (mm year$^{-1}$) for consistency across all water balance components. Shades indicate the standard deviation.

Surface evaporation (Fig. 5a) is directly tied to precipitation events, as it represents the immediate return of water to the atmosphere. Consequently, while being spatially consistent with the overall evaporation pattern (Fig. 4b), its distribution mirrors

that of rainfall (Fig. 4a). Thus, highest evaporation rates occurring directly from the surface are observed in high-altitude eastern sub-basins during the wet season (DJF). As the dry season progresses (MAM-JJA), surface evaporation declines sharply, as there is little to no remaining available water at the surface to evaporate. This reinforces the short-lived nature of surface evaporation, which primarily occurs immediately following rainfall events, rather than as a sustained water loss mechanism.

Unsaturated zone evaporation (Fig. 5b) represents the loss of soil moisture to the atmosphere before it can percolate into the groundwater reservoir. This process plays a critical role in regulating groundwater recharge, as it directly competes for moisture that could otherwise infiltrate to the groundwater reservoir. Unlike evaporation from water available at the surface, which closely follows precipitation patterns, evaporative losses from the unsaturated zone exhibit a ∼1-month lag with respect to surface evaporation. After the wet season ends, surface water availability declines, but residual soil moisture remains. With reduced cloud cover, increased solar radiation and drier atmospheric conditions, the evaporative demand is enhanced during March (Fig. 4b). During this transition period, due to lack of rainfall, soil moisture sustains evaporation while also sustaining groundwater recharge. Note the similarity in the annual cycle when comparing total evaporation and evaporation from the unsaturated zone, in particular the dynamic observed between January and March (Figs. 4b and 5b). The vadose zone therefore plays a crucial role in the system's water balance as it distributes moisture from rainfall events and split it between evaporation and groundwater recharge. By early winter, however, soil moisture becomes nearly depleted, which limits further percolation to the saturated zone during the second half of the year, while evaporation increases (Figs. 4b and 4c).

Groundwater evaporation (Fig. 5c) represents the final stage of evaporative losses, allowed in our model to occur only in areas where the water table is shallow enough to enable capillary rise (i.e., the active zone (Blin and Suárez, 2023)). Unlike surface and unsaturated zone evaporation, which are directly influenced by precipitation, groundwater evaporation displays the opposite behaviour and increases towards the end of year. During summer (DJF) and autumn (MAM) there is a relative small contribution from groundwater evaporation, but from winter on (JJA-SON) there is a sharp increase consistent with the rapid rise observed in total evaporation (Fig. 4b). Groundwater evaporation therefore responds to the atmospheric demand, reaching maximum values in spring (SON) that are comparable to those observed in surface evaporation during the rainy season (Fig. 5a). This dynamic reinforces the idea that, when no other moisture sources remain, groundwater becomes the primary supplier of evaporative fluxes. Consequently, water stored in the ground is primarily lost to the atmosphere rather than recharged during spring. In particular, we recognise the relevance of the salt flat nucleus sub-basin in sustaining the evaporative demands, being one of the main discharge mechanisms in the Salar del Huasco basin (Lobos-Roco et al., 2021). This is expected as the salt flat nucleus sub-basin is composed of wetlands and a shallow, salty lagoon that provide year-round water availability at the surface.

It is important to remember that these groundwater evaporation estimates represent the amount of water that the system needs for evaporation that is not supplied by direct precipitation and, as a result, is sourced from the aquifers. It is therefore a simplified back-trajectory analysis to identify its origin and it does not necessarily mean that evaporation happens directly from the water table. This can only happen in places like the main wetland where the water table is continuously exposed to the atmosphere in the form of a shallow lagoon. In other areas, this groundwater-sourced evaporation can be evaporated through capillary rise or transpiration from vegetation, which are processes we are not including in our model. To estimate the actual amount of water that is lost directly from the aquifer, information on groundwater levels and extinction depths are needed,

for which physically-based groundwater models are required (e.g., Blin and Suárez (2023), Diouf et al. (2020), Carroll et al. (2015)).

350  Since we recognised the salt flat nucleus sub-basin as an important evaporative ecosystem, we evaluated the role of groundwater flows in sustaining the observed high evaporation rates through a water balance analysis (Eq. 1, Fig. 2c). Figure 6a illustrates daily timeseries for precipitation, runoff, surface storage, total evaporation, base flow, and deep percolation. Notably, evaporation dominates the outflows throughout most of the record, with only brief periods during the rainy season (DJF) when inflows momentarily exceed evaporative losses. Because the salt flat lies at the lowest point of the basin and receives mini-

355 mal direct precipitation (Fig. 4a), such exceedances require either intense local rainfall or surface flow (runoff) from adjacent sub-basins. This suggests that, outside the rainy season, precipitation alone cannot sustain evaporation and, instead, it must be sustained by groundwater, as observed in Fig. 5c.

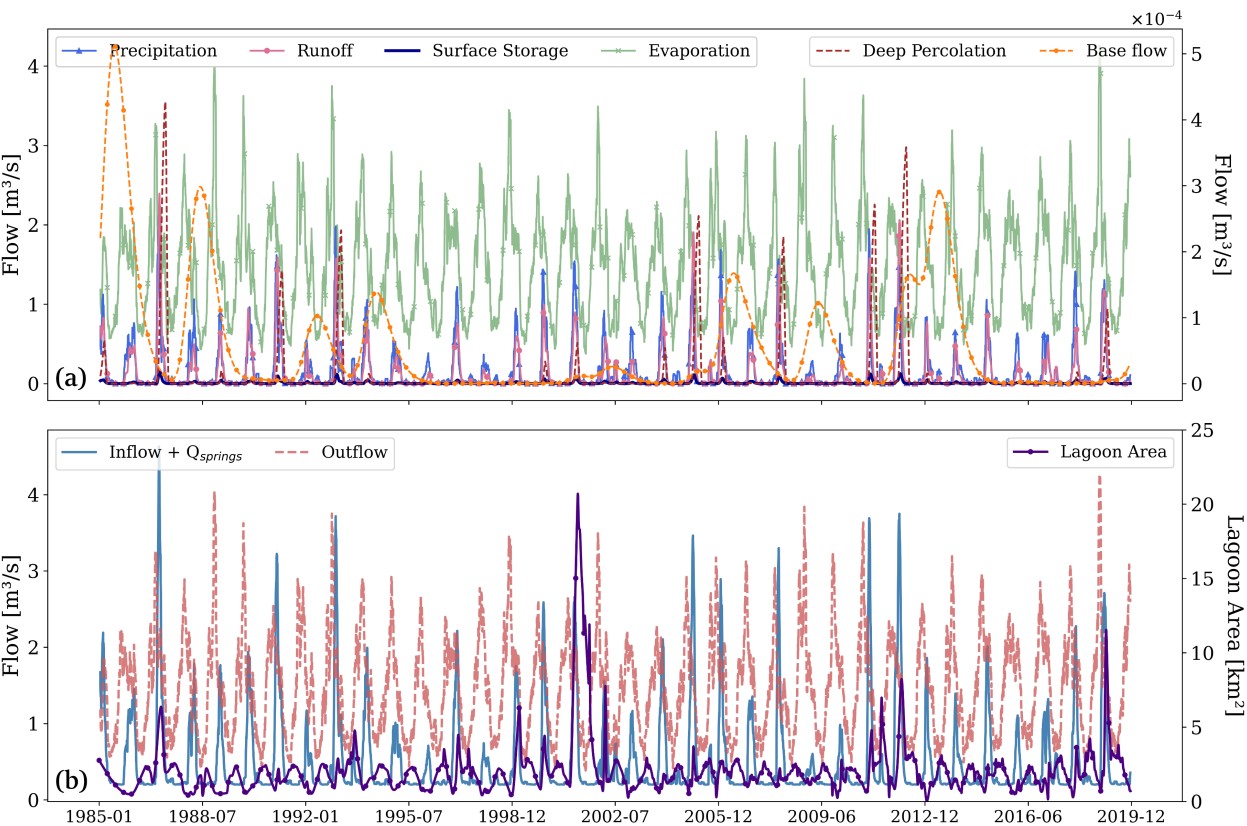

**Figure 6.** (a) Water balance components for the salt flat nucleus sub-basin, showing daily timeseries in $m^3s^{-1}$ of (left axis) precipitation, runoff, surface storage, total evaporation, and (right axis) deep percolation and base flow, all derived from the rainfall-runoff model. (b) Total inflow (sum of precipitation, runoff and base flow) and total outflow (sum of total evaporation and deep percolation) in $m^3s^{-1}$. Springs discharge ($Q_{springs}$) is added to the inflow and assumed constant at $0.2$ $m^3s^{-1}$ according to DGA hydrogeological reports. Lagoon area from Landsat imagery is displayed in the right y-axis [$km^2$]. A 30-days running average is applied to the timeseries.

To evaluate how this inflow-outflow dynamic influences actual changes in the shallow, salty lagoon extension, Fig. 6b displays total inflow (blue line) and total outflow (dashed red line), along with the lagoon area (purple line). Notably, outflows significantly exceed inflows, reinforcing the conclusion that evaporation drives the ecosystem's water balance. Evaporation dominance in this sub-basin is also evident from the lagoon-area fluctuations, as they respond more to daily outflows than to inflows (Fig. 6b). The lagoon exhibits an inverse seasonal pattern compared to the total inflow, with minimum area during the summer rainy season when inflows are highest and maximum area during winter (JJA) when evaporation is lowest (see also Fig. 1c). This indicates that the lagoon does not expand directly in response to rainfall events, but instead grows when evaporative losses are minimal, highlighting that evaporation is the dominant driver of the lagoon extent. This behaviour also suggests that groundwater plays a key role in sustaining the lagoon water levels during dry periods allowing a delayed hydrological response between precipitation and recharge. This is supported by the correlation analysis (Sect. 3.1), which shows that recharge peaks approximately one month after the rainy season, and continues to contribute for up to four months, extending its influence into early winter. This gradual accumulation of groundwater recharge likely explain the progressive increase in lagoon area from late summer into winter, when it reaches its annual maximum. The lagoon then begins to shrink as evaporation increases in spring (SON).

Note that extreme rainfall events are an exception to this typical annual cycle. During years such as 1987, 2001, 2011, and 2019, sudden expansions of lagoon area are observed during summer (Fig. 6b). This is likely due to large-magnitude storm events that generate rapid surface runoff and temporary increases in surface storage (see also Fig. 1 in Lobos-Roco et al. (2021)). This dynamic response to precipitation and groundwater flows align with permanent and temporary (or transitional) lagoons observed in the region, where permanent lagoons (approximately 2 km$^2$ in the Salar del Huasco basin) are sustained primarily by groundwater dynamics, while temporary lagoons emerge following extreme precipitation events (Boutt et al., 2016; Acosta, 2004). Such episodic surface storage pulses can be seen in Fig. 6a, for instance, during 1987 and 2012.

To further examine the relationship between the hydrological variables and lagoon area we estimated orthogonal regressions on monthly averages, distinguishing between magnitude (i.e., the longer-term or absolute area of the lagoon) and fluctuations (the shorter-term ups and downs). Table 1 shows that total inflow has a stronger correlation with lagoon-area fluctuations ($R^2$ = 0.51) than with lagoon-area magnitude ($R^2$ = 0.01). These results align with the idea that abrupt lagoon expansions respond to rainfall and runoff events, highlighting their influence on the lagoon short-term variability rather than long-term stability. Total outflow, instead, has a stronger correlation with area magnitude ($R^2$ = 0.66). This is consistent with total evaporation ($R^2$ = 0.66) driving the long-term water-level baseline that controls the lagoon area. Groundwater-related variables, particularly groundwater evaporation ($R^2$ = 0.56) and base flow ($R^2$ = 0.61), explain this strong correlation between outflow and lagoon area magnitude. These results underscore the role of subsurface flows in controlling overall water levels. In contrast, surface variables, both inflows and outflows, explain more of the short-term fluctuations, including precipitation ($R^2$ = 0.53), runoff ($R^2$ = 0.47), surface storage ($R^2$ = 0.43), and surface evaporation ($R^2$ = 0.44). These results also highlight the episodic influence of storm events, and their associated surface response, on the lagoon size. In short, these findings reveal two distinct timescales in lagoon behaviour, with short-term expansions governed by surface inflows and outflows, and longer-term persistence controlled by groundwater discharge and groundwater evaporation mechanisms.

**Table 1.** Orthogonal regression $R^2$ values for Lagoon Area Magnitude and Lagoon Area Fluctuations

| Variable | Magnitude $R^2$ | Fluctuations $R^2$ |
|---|---|---|
| $\Delta$Storage | 0.45 | 0.46 |
| Inflow | 0.01 | **0.51** |
| Outflow | **0.66** | 0.04 |
| **Surface Inflows** | | |
| Precipitation | 0.03 | 0.53 |
| Runoff | 0.00 | 0.47 |
| Surface Storage | 0.01 | 0.43 |
| **Surface Outflows** | | |
| Surface Evaporation | 0.01 | 0.44 |
| Unsat. Zone Evaporation | 0.02 | 0.18 |
| **Groundwater Outflows** | | |
| Groundwater Evaporation | 0.56 | 0.31 |
| Total Evaporation | 0.66 | 0.04 |
| Deep Percolation | 0.10 | 0.02 |
| Base flow | 0.61 | 0.05 |

An additional noteworthy feature is that springs flow remain relatively stable over time, according to historical DGA records, even with such strong seasonal changes in the atmospheric variables. This constancy suggests a dynamic groundwater system that can regulate its outflow and buffer the lagoon and nearby wetlands against extreme fluctuations. The underlying groundwater reservoir most likely stores water during wet years (via deep percolation and recharge) and gradually releases it during dry years, sustaining terminal wetlands and lagoon water levels as described by de la Fuente et al. (2021). This suggests that springs discharge and the groundwater reservoir play a key role in controlling the water balance of the shallow lagoons and wetlands.

395

## 3.3 Questioning the Endorheic Paradigm

From the previous section, we observed that outflows considerably exceed the inflows in the salt nucleus sub-basin, suggesting that evaporation causes a significant negative water balance or that the ecosystem is sustained by groundwater inflow, or other processes, currently not accounted. Given the limited local recharge due to minimal precipitation and the lack of evidence of the lagoon shrinking, the alternative of missing inflows warrants further investigation. The magnitude of this imbalance also opens the possibility that the assumption of the basin's endorheic nature may be invalid. Otherwise, it means that the basin may be suffering a net water loss.

To discuss this further, we evaluate the water balance at the scale of the entire Salar del Huasco basin. Figure 7 provides a conceptual overview of the long-term annual mean water balance for the Salar del Huasco, as derived from our 40-year analysis. These results reveal a pronounced imbalance of $\sim$1.5 in which evaporation (10.35 m$^3$s$^{-1}$, 230 mm year$^{-1}$) considerably exceeds precipitation (6.70 m$^3$s$^{-1}$, 145 mm year$^{-1}$). This might be associated with certain limitations that we must acknowledge.

First, there are unmodelled processes, such as snow and frozen soil melting that have been observed in-situ during spring. These processes can represent additional water sources that sustain evaporation outside the rainy season. However, contributions from snow and frozen soil melt are not significant enough to counteract the large negative imbalance given by evaporation.

Second, our estimates are subject to uncertainties associated with satellite-derived input data and our interpolation approach (Appendix B). Although local bias corrections were applied using available observations, residual errors persist and inevitably propagate into the water balance analysis. At the station level, precipitation remains systematically overestimated, with percent bias values ranging from -0.3% to +26%, particularly at lower-elevation sites (Table B1, Fig. B2). When aggregated at the basin scale, the median signed bias is -10%. Based on this analysis and previous assessments of CR2MET performance, we adopt $\pm$10% as a representative uncertainty bound for precipitation. For evaporation, while the temporal behavior of EEFLUX was validated against ground-based pan observations at a representative site in the salt flat nucleus (Fig. B5), the spatial coverage of ground data was insufficient to directly assess uncertainty across the basin. We therefore adopt a conservative uncertainty bound of $\pm$20%, consistent with validation studies of EEFLUX/METRIC in arid and high-elevation regions reporting typical errors of 5–20% (Nisa et al., 2021; Wasti, 2020; Lima et al., 2020; Madugundu et al., 2017). Even under the most favorable case, precipitation increased by 10% and evaporation decreased by 20%, the imbalance persists. Moreover, given that precipitation is more likely overestimated, the reported deficit is likely conservative, and the true imbalance may be larger. Thus, we do not attribute the imbalance primarily to data uncertainty.

One hypothesis explaining the imbalance is that this deficit arises from differences in the timescales at which key processes operate. Since most evaporation originates from groundwater, which moves more slowly than surface or atmospheric systems, the total evaporative loss over the 40-year analysis period could exceed the precipitation that falls within the basin. However, given that four decades is a considerable timescale, it is unlikely that this factor alone can account for such a pronounced discrepancy. A second hypothesis is that small declines in groundwater-reservoir storage could account for the imbalance, since even a few millimeters of water-level reduction in a basin of this size ($\sim$ 1500 km$^2$) would produce significant volumetric

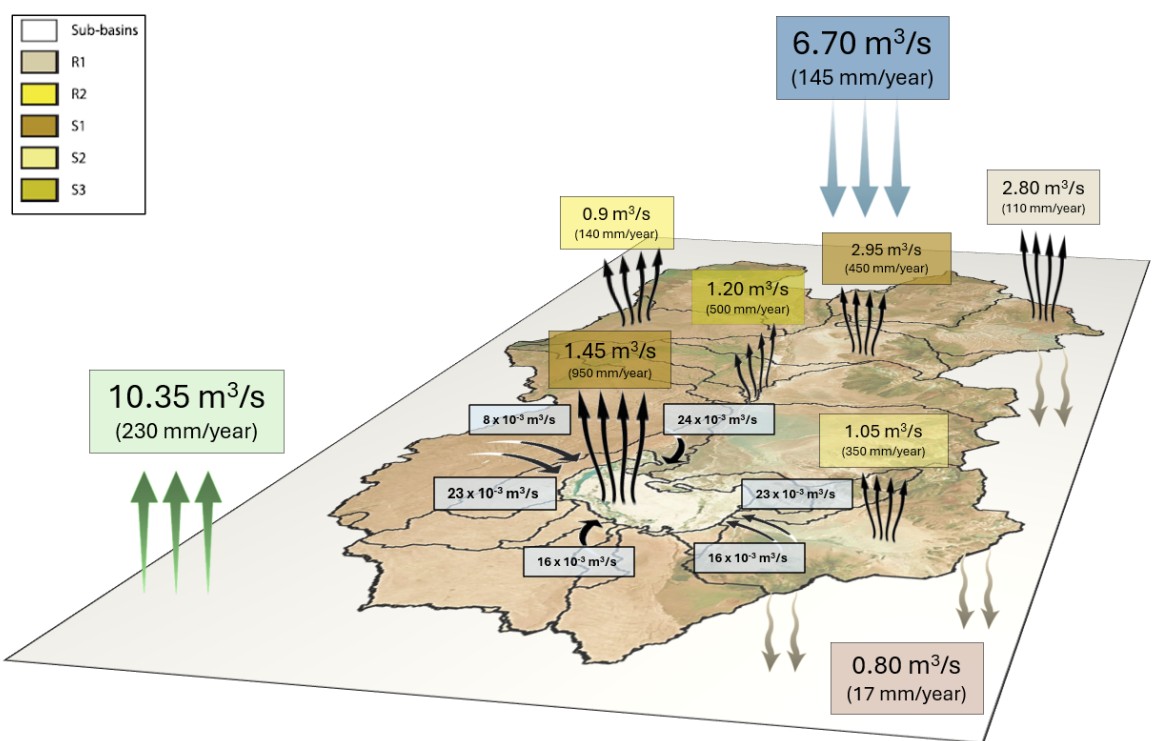

**Figure 7.** Schematic representation of long-term mean annual flows from the Salar del Huasco, Chile, in $m^3 s^{-1}$ and in parenthesis mm year$^{-1}$. Total evaporation is presented in green, total precipitation in blue and total groundwater recharge in brown. Springs are displayed in light blue. The rest represent evaporative losses from each Hydrological Response Unit (see Fig. 2b). Evaporation from the salt flat nucleus basin is separated from the rest S1-type, as it is the main discharge point per unit area of the basin (950 mm year$^{-1}$).

changes. The estimated imbalance of -85 mm year$^{-1}$ would imply a water level decrease of $\sim$ 3.5 meters over 40 years, suggesting a significant net water loss from the basin. Yet, multiple wells across the Salar del Huasco show no evidence of such long-term groundwater-level decline (see Blin et al. (2022)). Although soil in these arid regions is gradually drying as a result of increasing evaporation rates (Lobos-Roco et al., 2022; Satgé et al., 2017; Valdivia et al., 2013), it is unlikely that moisture depletion in the vadose zone would greatly contribute to the imbalance. In fact, if soil moisture was being depleted at such pronounced rate, one would also expect a measurable decline in groundwater levels, particularly in areas where evaporation is supported by capillary rise from shallow water tables. In areas with deeper water tables, significant soil moisture depletion is also questionable, as evaporation is generally limited and cannot dry deeper soil layers. Given this, we question if the pronounced imbalance is more plausibly explained by unaccounted groundwater inflows. This, in turn, raises the possibility that groundwater may be entering the basin from beyond its currently defined hydrogeological boundaries.

The above leaves us to discuss our main hypothesis, which is that although the Salar del Huasco basin is classified as endorheic at the surface, it may be hydrogeologically open. This would suggest that the topographic watershed does not match

the actual extent of the groundwater system. This kind of mismatch is relatively common in arid and semi-arid regions, where inter-basin groundwater flows exist despite appearing closed based on surface topography (e.g., Ordos and Heihe River Basins in North China (Zhang et al., 2025; Li et al., 2018), Verlorenvlei Basin in South Africa (Welham et al., 2025), Northern Sahara basin in Morocco (Alcalá et al., 2018), and Great Basin in California, United States (Belcher et al., 2009)). Similar cases have been also documented in the Andes region. Alvarez-Campos et al. (2022) showed that Laguna Salinas in southern Peru, although hydrologically closed, supplies groundwater to springs located beyond its surface basin, suggesting a wider hydrogeological catchment. Groundwater flow between closed basins in volcanic terrains in the Chilean Altiplano has also been previously documented, for instance, in the Salar de Michincha and Salar de Coposa basins (Montgomery et al., 2003), as well as between Laguna Tuyajto and adjacent basins (Herrera et al., 2016). Moreover, Moran et al. (2019) found that groundwater recharge in the Salar de Atacama originates in higher-elevation basins beyond its topographic boundary, with subsurface inflows playing a key role in sustaining the system's water balance. In the Atacama Desert, the groundwater boundaries of the Loa Basin are not well defined and are also believed to extend eastward beyond the surface catchment, possibly beneath the Altiplano Plateau (Jordan et al., 2015). The case of the Silala River further demonstrates how groundwater can contribute to surface flows even across international boundaries. Both Bolivia and Chile recognise a groundwater catchment that is significantly larger than the topographic catchment, where regional groundwater flow sustains the river and influences its transboundary hydrology and water use (Peach and Taylor, 2024; Muñoz et al., 2024; Wheater et al., 2024). Although the Salar del Huasco basin has previously been considered hydrogeologically closed (Uribe et al., 2015; Blin et al., 2022), the pronounced imbalance identified from this study, along with the examples discussed here, support the option of re-evaluating this assumption and investigating the potential for external subsurface contributions to the Salar del Huasco (Cifuentes et al., 2018; Moya and Scheihing, 2018). To this end, groundwater observations are essential to better constrain the hydrological model and groundwater recharge estimates. Further validation of satellite-derived inputs against local observations across the basin, particularly for evaporation and air temperature, is also strongly recommended to enhance the robustness of the water balance assessment. In addition, a more integrated and robust methodological approach is recommended to quantify subsurface storage changes and to estimate how large the groundwater catchment would need to be to explain the observed water balance of the basin. Overlooking such groundwater inflows may lead to inaccurate water budgets and potentially unsustainable water management decisions (Hammond and Han, 2006; Le Moine et al., 2007).

In summary, the water imbalance could result from one or more of several factors: (1) inaccuracies in satellite-derived estimates, (2) unrecognised declines in groundwater storage, (3) unaccounted snowmelt or frozen-soil contributions, or, most critically, (4) external groundwater inflows that challenge existing assumptions about the basin's hydrogeological limits. Given the magnitude of the imbalance and supporting evidence from literature, the most likely explanation is that the Salar del Huasco basin might not be hydrogeologically closed. Overall, these findings point to groundwater redistribution, rather than direct precipitation or runoff, as the critical driver of the basin's evaporative demand and water balance. This underscores the importance of groundwater in hydrological balances in arid regions, especially in those with strong seasonal precipitation as the Altiplano.

 **4    Conclusions**

This study investigated the spatiotemporal variability of groundwater recharge and upwelling in the Salar del Huasco, focusing on its interaction with precipitation and evaporation dynamics. Using a modified rainfall-runoff model coupled with satellite-derived datasets, we quantified the key fluxes regulating the basin's water balance and demonstrated that evaporation is the dominant water loss mechanism and main driver of the water balance. Results showed that groundwater recharge responds to the combined seasonal evolution of precipitation and evaporation. It occurs in high-elevated sub-basins during summer when precipitation exceeds evaporation, but primarily during autumn when evaporation is lower compared to the austral summer. From there on, recharge diminishes to minimum levels due to the lack of precipitation and the increasing atmospheric demand for evaporation, which is mainly met by groundwater. At that point, subsurface storage plays a critical role in regulating water availability by balancing evaporation with groundwater recharge. This groundwater dynamic is also observed over longer timescales, as the basin's water balance relies on episodic wet years (every 20 - 30 years) that recharge the aquifers. The stored groundwater is then gradually released, sustaining evaporation and supporting unique ecosystems during dry periods.

Beyond these dynamics, our analysis highlights a fundamental issue: the observed water balance cannot be reconciled under the traditional endorheic paradigm. This paradigm assumes that endorheic basins are hydrogeologically closed systems in which atmospheric inputs are balanced internally by evaporation and storage changes. However, our results suggest that groundwater inflows from outside the topographic basin boundaries are needed to explain the observed high evaporation rates, therefore challenging current assumptions about the endorheic nature of the Salar del Huasco basin. Future research should further investigate this possibility and incorporate fully coupled models that integrate both surface and groundwater processes, particularly in areas with shallow water tables. Also, incorporating groundwater-level or tracer data, where available, is strongly recommended to provide independent validation of the recharge estimates, which in this study should be considered as first-order approximations. Furthermore, refining the model structure to reduce parameterization, as demonstrated in the Silala basin (Yáñez-Morroni et al., 2024a, b), would help minimize parameter uncertainty and enhance the robustness of groundwater applications. Finally, validating satellite-derived inputs against additional local observations is essential to strengthen the reliability of the water balance assessment. Representing and understanding these processes is crucial for water resources management and environmental protection in the Altiplano, where climate change is posing increasing risks to water availability and to the unique ecosystems that strongly control the hydrological response of these arid basins.

**Appendix A:  Modified Rainfall-Runoff model**

The modified rainfall-runoff model calibration was conducted using daily streamflow data from the Collacagua River and Piga River at Collacagua (DGA, Chile) for the analysed period (1980–2019). Uribe et al. (2015) performed a sensitivity analysis in which they identified five key parameters as the most influential on model performance: Percentage of soil infiltration ($\alpha$), Maximum infiltration rate between the surface and the first soil layer (KDS), Storage capacity in the root zone (SVR), Storage capacity in the intermediate capillary zone (SVS), Storage capacity in the saturated zone (CASV). Only these five parameters

were calibrated, while the remaining 10 parameters were assigned values from previous regional studies (DIHA-PUC, 2009; Blin et al., 2022). Figure A1 shows the calibration results for Piga and Collacagua River.

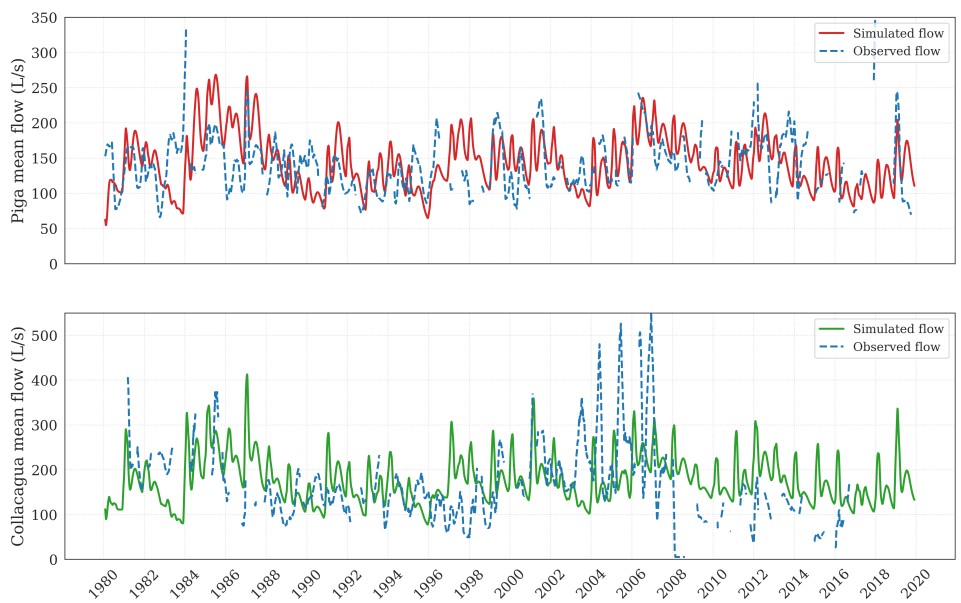

**Figure A1.** Observed vs. simulated daily stream flows at the Piga River and Collacagua River in L/s.

Limitations must be considered when interpreting the results of the modified rainfall-runoff model. First, evaporation during the dry season is not dynamically simulated but rather prescribed based on satellite-derived estimates. This simplification can lead to under- or overestimation of seasonal water losses and, consequently, affect recharge estimates, particularly during low-flow periods. Second, the model relies on satellite-derived precipitation, temperature, and evaporation inputs that, despite being bias-corrected, retain residual uncertainties, especially in regions with sparse ground observations. These uncertainties propagate through the model and may influence the accuracy of water balance components. Third, key high-altitude hydrological processes such as snow accumulation, melt runoff, and freeze-thaw soil dynamics are not represented. While not dominant, these processes have been observed in the Salar del Huasco and may influence the timing and magnitude of infiltration and recharge. Additionally, calibration is based on observed streamflows that are subject to measurement uncertainty, particularly during periods of low or ephemeral flow, which can affect model performance assessments. Lastly, the use of 15 calibrated parameters increases the risk of equifinality, where different parameter combinations yield similar model performance but diverge in internal process representation. This limits confidence in the uniqueness and robustness of the simulated recharge. Following recent recommendations (e.g., Yáñez-Morroni et al. (2024a, b)), future work should aim to reduce model complexity and improve parameter identifiability through more parsimonious formulations. Despite these limitations, the model provides a spatially distributed, process-oriented framework that enables exploration of the dominant hydrological drivers in the Salar del

Huasco basin. The results should be interpreted as indicative, with future refinements benefiting from improved observational data and enhanced model structure.

## Appendix B: Satellite-derived data

Remote sensing and reanalysis data are particularly valuable in the Altiplano, where the harsh climate and remote terrain restrict the availability of ground-based observations. In this study, we integrated gridded datasets into the Uribe et al. (2015) rainfall-runoff model to better characterise the hydrometeorological input data. In the following sections we detail the datasets used for precipitation and air temperature, as well as for evaporation. Details regarding daily interpolation and bias correction approaches to fit the data to Salar del Huasco local conditions are also explained.

### B1  Precipitation and Air Temperature

In order to capture the spatiotemporal variability of precipitation and air temperature in the Salar del Huasco basin, we employed the Center for Climate and Resilience Research Meteorological (CR2MET) data version 2.0 (daily resolution, 0.05° latitude-longitude) spanning 1980 to 2019. The CR2MET products, originally developed for continental Chile, are based on statistical models calibrated against quality-controlled in situ records (Boisier et al., 2018). They integrate ECMWF ERA5 reanalysis data, topographic parameters, and MODIS land surface temperature estimates (Boisier et al., 2018).

Since we required an accurate description of precipitation and air temperature, we proposed a machine learning based approach for bias correction using ground-based observations from the Chilean National Water Division (Dirección General de Aguas, DGA). Figure B1 summarises the workflow for the bias-correction procedure. First, we performed a bicubic downscaling process from ∼5 km to ∼1 km to achieve higher spatial resolution in the CR2MET gridded fields. Next, common dates between the datasets and DGA observations were identified for each station to enable comparison. Since few DGA stations were available for the Salar del Huasco basin (Fig. B2), the extracted data were then transformed into "synthetic timeseries" by concatenating information from multiple stations. With this approach we ensured a structured and larger input for model training. Next, we defined the feature input data for the machine learning model, which consisted of the concatenated CR2MET timeseries, along with temporal and spatial predictors, namely month, day, latitude, and longitude. The target variable was the corresponding DGA observational data, creating a supervised learning problem in which the model learned the mapping function between the raw CR2MET values and ground-truth observations. Subsequently, a data-driven correction approach was implemented by training a Random Forest (RF) regression model on the compiled dataset to capture and correct systematic biases ($\varepsilon$) in the prediction. The trained model was then applied to adjust CR2MET values according to the learned relationships, producing bias-corrected estimates based on the functional transformation:

$$CR2MET_{corrected} = \phi_{RF}(CR2MET, month, day, latitude, longitude) + \varepsilon \qquad \text{(B1)}$$

where $\varepsilon$ is the bias in the RF model.

The final prediction is therefore the prediction from the first model corrected by the data-driven bias approach. Following
560 the correction, a postprocessing step was carried out to ensure physical consistency in the precipitation estimates by imposing
a non-negativity constraint (i.e., setting negative rainfall values to zero).

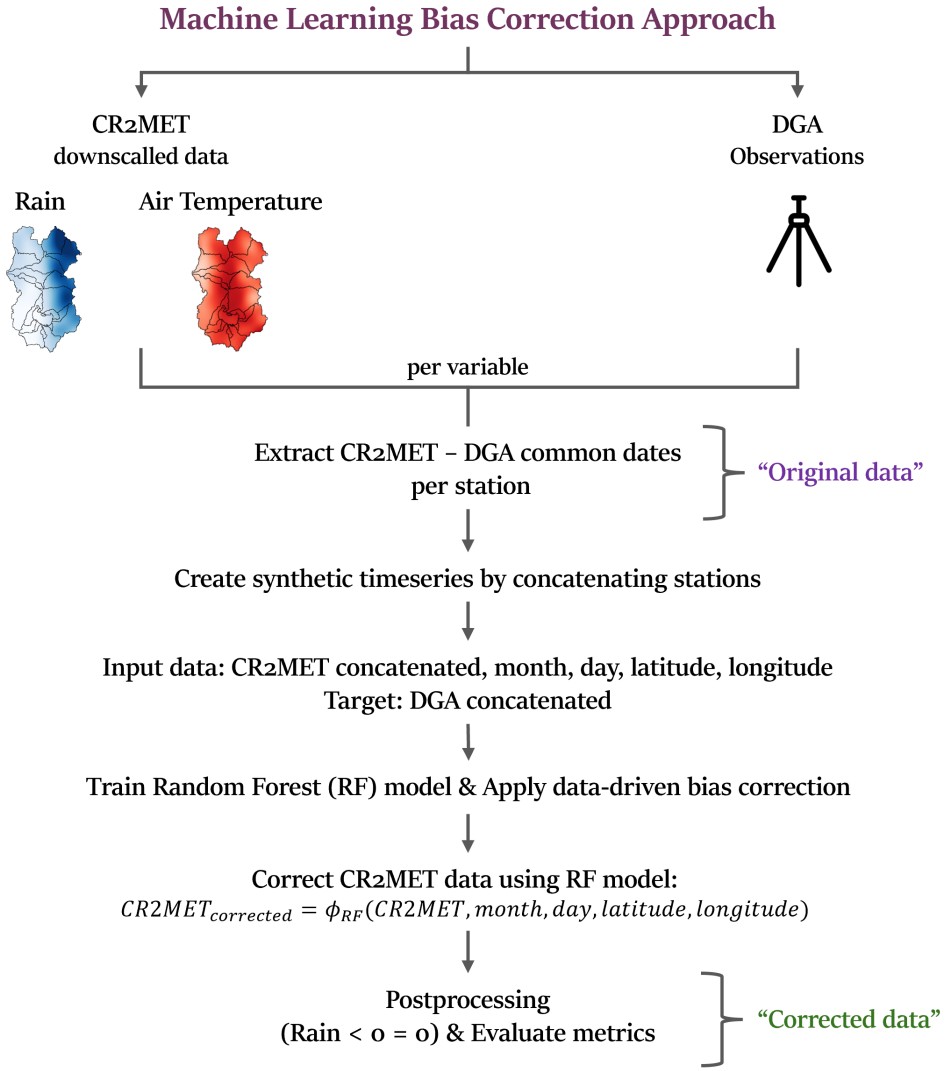

**Figure B1.** Schematic representation of the machine learning-based bias correction approach for CR2MET precipitation and air temperature data using DGA observations.

Tables B1 and B2 summarize the metadata of the precipitation and temperature stations used in the bias correction, together with performance metrics that compare ground-based observations against the corrected CR2MET datasets.

**Table B1.** Summary of validation metrics for daily precipitation at DGA meteorological stations used for CR2MET bias correction. The colors in parentheses correspond to station markers in Fig. B2. PBIAS represents the percent bias computed for all days with recorded precipitation > 1 mm, RMSE is the root mean square error, and $R^2$ is the coefficient of determination. The basin median signed bias is derived from aggregating all station records into a single time series.

| Station | Lat/Lon | Elevation (m asl) | Record length | PBIAS (%) | RMSE (mm day$^{-1}$) | $R^2$ |
|---|---|---|---|---|---|---|
| Coyacagua (green) | -20.04°/-68.83° | 4013 | 14326 | 22.4 | 1.05 | 0.67 |
| Río Collacagua (red) | -20.10°/-68.84° | 3853 | 612 | 11.4 | 0.56 | 0.86 |
| Diablo Marca (orange) | -20.10°/-68.97° | 4585 | 1397 | 0.7 | 0.93 | 0.69 |
| Sillillica (purple) | -20.17°/-68.74° | 3840 | 1427 | -0.3 | 1.03 | 0.79 |
| Salar Huasco (black) | -20.28°/-68.89° | 3800 | 457 | 26.1 | 0.81 | 0.49 |
| Altos del Huasco (brown) | -20.32°/-68.89° | 4044 | 1449 | 2.6 | 0.62 | 0.76 |
| Basin median signed bias (%) | -9.1 | | | | | |

**Table B2.** Summary of validation metrics for daily air temperature at DGA meteorological stations used for CR2MET bias correction. The colors in parentheses correspond to station markers in Fig. B2. PBIAS represents the percent bias, RMSE is the root mean square error, and $R^2$ is the coefficient of determination. The basin median signed bias is derived from aggregating all station records into a single time series.

| Station | Lat/Lon | Elevation (m asl) | Record length | PBIAS (%) | RMSE (°C) | $R^2$ |
|---|---|---|---|---|---|---|
| Coyacagua (green) | -20.04°/-68.83° | 4013 | 11383 | -0.2 | 1.52 | 0.86 |
| Salar Huasco (black) | -20.28°/-68.89° | 3800 | 300 | 3.9 | 1.24 | 0.88 |
| Basin median signed bias (%) | -0.9 | | | | | |

Figure B2 presents the comparison between the observations and the original and corrected CR2MET data for Salar del Huasco basin. The original data shows a systematic underestimation of high precipitation events, coupled with an overestimation of low-precipitation conditions. These findings are consistent with prior studies indicating that satellite-derived or reanalysis-based products often exhibit biases in mountainous or arid settings, owing to complex orographic effects, sparse station networks, and challenges in accurately representing convective storms (e.g., Baig et al. (2025), Alvarez-Garreton et al. (2018), Boushaki et al. (2009)). For air temperature, the original gridded dataset generally suggested a better fit to the observations compared to precipitation. However, some deviations persisted, especially under extreme temperature scenarios. Although each correction and downscaling step can introduce its own uncertainties, the correction procedure collectively improve the agreement of modelled outputs with ground-based observations for both precipitation and air temperature.

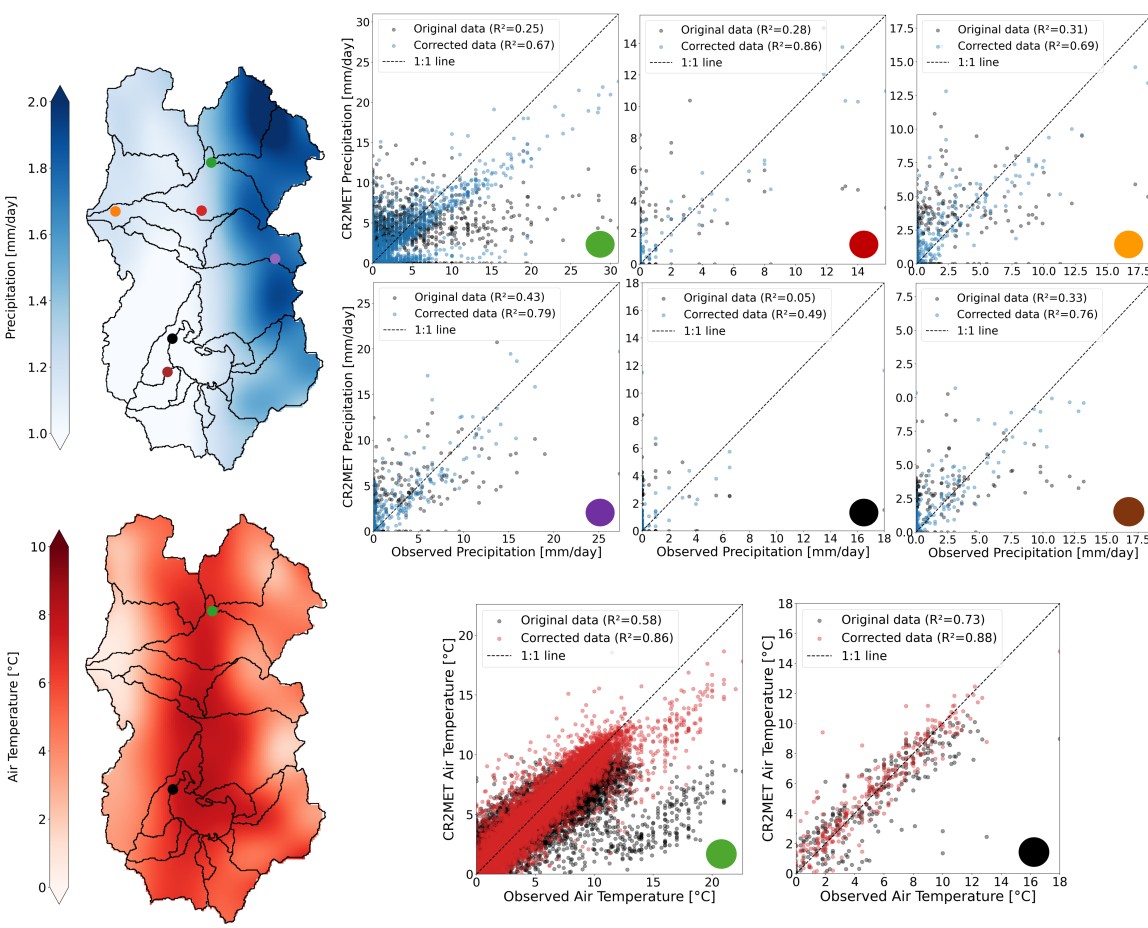

**Figure B2.** Original (black dots) and corrected CR2MET data (blue dots for precipitation and red dots for air temperature) for the Salar del Huasco basin. A machine learning bias correction approach was implemented (Fig. B1).

## B2 Evaporation

In contrast to variables like precipitation or temperature, evaporation cannot be directly measured by satellites. Instead, temperature-based evaporation models are commonly used, which implement thermal remote sensing to calculate sensible heat fluxes and estimate latent heat fluxes as a residual of the surface energy balance (e.g., Derardja et al. (2024), Zhang et al. (2016), Aryalekshmi et al. (2021)).

In this study, we implemented the Earth Engine Evapotranspiration Flux (EEFLUX) tool, which applies the Mapping Evapo-Transpiration at high Resolution with Internalized Calibration (METRIC) model to Landsat images to produce 30 m-resolution evaporation estimates every ~16 days (https://eeflux-level1.appspot.com/). This high spatial resolution is especially advantageous in arid mountainous regions such as the Altiplano, where localised evaporation processes (e.g., from wetlands and shallow lagoons) can vary significantly over short distances (Blin et al., 2022; Lobos-Roco et al., 2021). Blin and Suárez (2023) used EEFLUX data as observations to calibrate a hydrogeological model developed for the Salar del Huasco basin, finding good agreement for desert, water and salt surfaces. Good EEFLUX performance was also found for other basins in the Atacama Desert, such as Silala basin, Putana river wetland, and salt surfaces and wetlands in Arica and Parinacota Region (see Figs. 5 and 6 in Blin and Suárez (2023)). However, while EEFLUX provides an opportunity to capture finer-scale features, its reliance on meteorological inputs and satellite-based surface temperature introduces uncertainties, particularly under cloudy conditions or where complex terrain can alter the energy balance (Allen et al., 2007). These uncertainties need to be taken into account when using EEFLUX estimates.

Non-linear relationships between evaporation drivers (e.g., rainfall, soil moisture, temperature) make machine learning an attractive tool to estimate evaporation compared to linear approaches, such as multiple linear regressions. To address the temporal resolution gap created by the 16-day Landsat cycle, we therefore applied a machine learning approach using CR2MET corrected precipitation and air temperature data (see Sect. B1). With this approach we also ensure consistency between the gridded datasets that are integrated in the rainfall-runoff model. Our methodological flow is summarised in Fig. B3.

Initially, EEFLUX data per Hydrological Response Unit (HRU, see Fig. 2) were extracted as the reference dataset ("original data" in Fig. B3). To build a longer training dataset, "synthetic timeseries" were created by concatenating data across multiple basins for each HRU-type, allowing the model to capture spatial variability due to geological conditions while maintaining a structured temporal resolution. The input features for the machine learning model included bias-corrected precipitation and air temperature from CR2MET ($P_{corrected}$, $Ta_{corrected}$ from Sect. B1), along with temporal (month and day) and spatial features (latitude and longitude). The target variable was the observed EEFLUX at the concatenated HRU level. A Random Forest model was then trained separately for each HRU-type, enabling soil-type specific bias corrections and ensuring that geological patterns are respected (see model performance per HRU in Fig. B4). Next, the HRU-RF models were applied to each independent and corresponding HRU to obtain daily EEFLUX estimates ($R^2 \geq 0.75$). To mitigate the inherent bias introduced by machine learning models, a postprocessing step was then applied. First, physical constraints were imposed to ensure non-negative evaporation ($E$) estimates ($E = 0$ if $E < 0$) and coherence with air temperature data ($E = 0$ if $T_a < 0$) . Additionally, an "original data correction" step was implemented to ensure seasonal averages were not altered in the machine

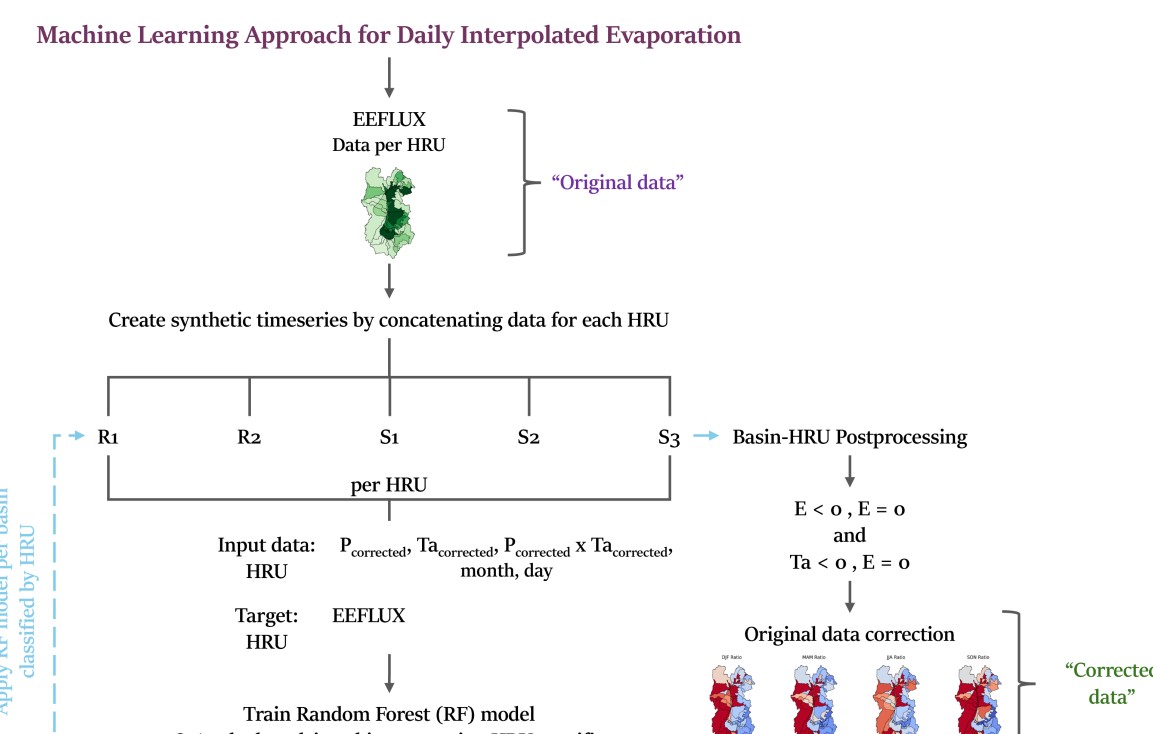

**Figure B3.** Schematic representation of the machine learning-based approach for EEFLUX daily interpolation.

learning process. For this, EEFLUX estimates were corrected by a ratio estimated from the initial dataset (temporal resolution $\sim$ 16 days) seasonal mean. This correction was essential to preserve the statistical consistency of the original data and prevent systematic overestimations, as machine learning models often struggle with accurately predicting values close to zero. Although EEFLUX's accuracy diminishes when cloud cover or mountainous topography disrupts satellite retrievals, this hybrid approach of high-resolution remote sensing and adaptive machine learning offers a powerful tool for capturing fine-scale evaporation patterns in data-scarce, arid environments, such as the Altiplano.

We assessed EEFLUX performance by comparing its actual evaporation estimates with pan-derived potential evaporation, adjusted using a 0.7 coefficient as recommended by DGA (2009). These observations were collected at a representative site within the salt flat nucleus sub-basin (GP Consultores, 2008), the area with the highest evaporative fluxes in the study site. As expected, observed potential evaporation exceeds EEFLUX-derived actual evaporation by 18% (Fig. B5), consistent with the differences between potential and actual evaporation. While this comparison is not suited for validating absolute magnitudes, it provides useful insight into the temporal behavior of EEFLUX, which reproduces both seasonal and daily variability captured by the pan observations well (Fig. B5). Given the limited spatial and temporal coverage of these observations and the nature

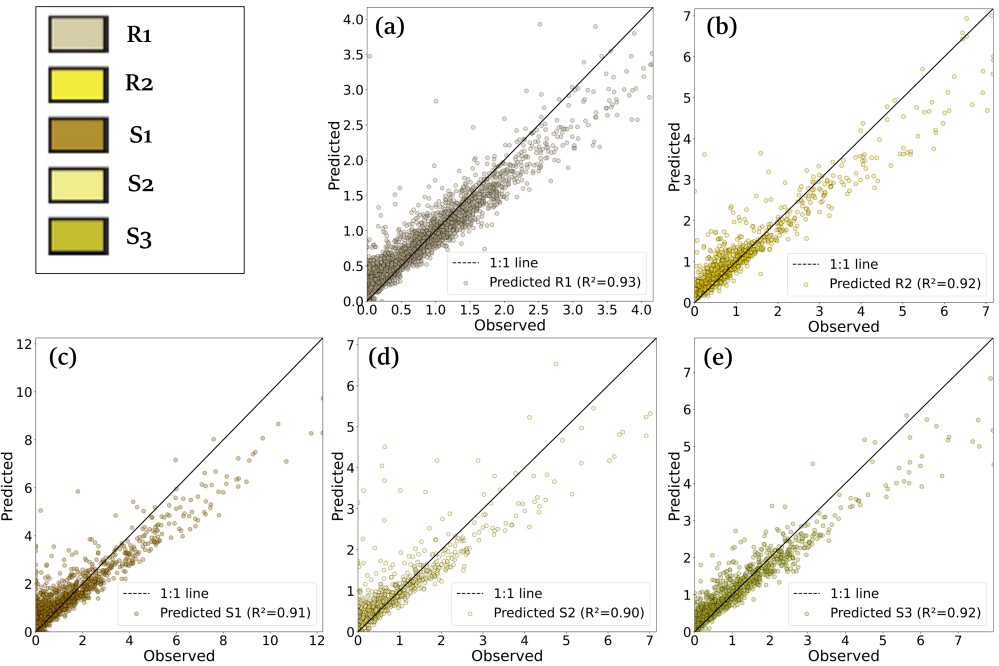

**Figure B4.** Training results for daily interpolating EEFLUX evaporation data for each Hydrological Response Unit: (a) R1, (b) R2, (c) S1, (d) S2, and (e) S3.

of the metric used, this comparison supports the plausibility of EEFLUX estimates but is not considered a formal validation. Thus, we consider a conservative $\pm 20\%$ uncertainty bound based on published studies (e.g., Nisa et al. (2021); Wasti (2020); Lima et al. (2020); Madugundu et al. (2017)) when propagating evaporation errors into the water balance analysis.

## B3 Lagoon Area

The lagoon's surface area was estimated using Landsat 5 and Landsat 8 Collection 2 Level 2 Surface Reflectance products (30 m spatial resolution, $\sim 16$ days temporal resolution) through a systematic remote sensing approach. Initial preprocessing involved applying standard scaling factors to convert digital numbers to surface reflectance values and spatiotemporal filtering. Images with cloud cover exceeding 10% were discarded to minimise atmospheric interference. The Modified Normalized Difference Water Index (MNDWI) was applied to enhance water body detection, using empirical thresholds of $\geq 0.4$ for Landsat 5 and $\geq 0.5$ for Landsat 8. In addition, a Near-Infrared (NIR) band threshold of $\leq 0.3$ was used to filter out potential snow pixels. The thresholds were determined by careful inspection using optical images from multiple dates. The resulting binary water classification allowed for the computation of total lagoon surface area.

To generate a continuous daily time series, a piecewise cubic interpolation method was applied, ensuring smooth and realistic reconstruction of short-term variations while maintaining long-term trends. This approach effectively handled irregular

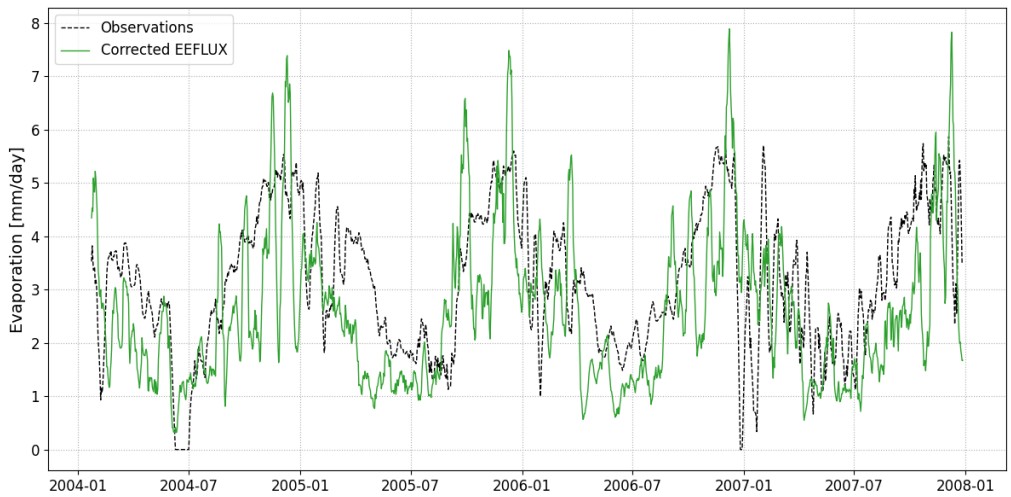

**Figure B5.** Comparison of EEFLUX-derived daily evaporation at the salt nucleus sub-basin (green) with ground-based observations of potential evaporation (black) at a representative location in the salt flat nucleus sub-basin (GP Consultores, 2008). Both time series have been smoothed using a 10-day rolling average to highlight temporal variability.

temporal gaps due to cloud cover and due to temporal resolution ($\sim$ 16 days). Note that, since our goal was to analyse how
the lagoon's area fluctuates in response to precipitation variability, we did not use the machine learning approach proposed for evaporation (see previous section), as this would artificially introduce the relationship. The estimated lagoon extent was validated against MODIS-derived daily surface water observations, demonstrating similar temporal variability. We used the MODIS Terra Surface Reflectance dataset (MOD09GA v061 available from 2000 onwards) and applied the MNDWI with a threshold of $\geq$0.3, following a similar methodology to that used with the Landsat collections. Differences in absolute magnitude
were attributed to the higher spatial resolution of Landsat, which provides more detailed water extent estimations.

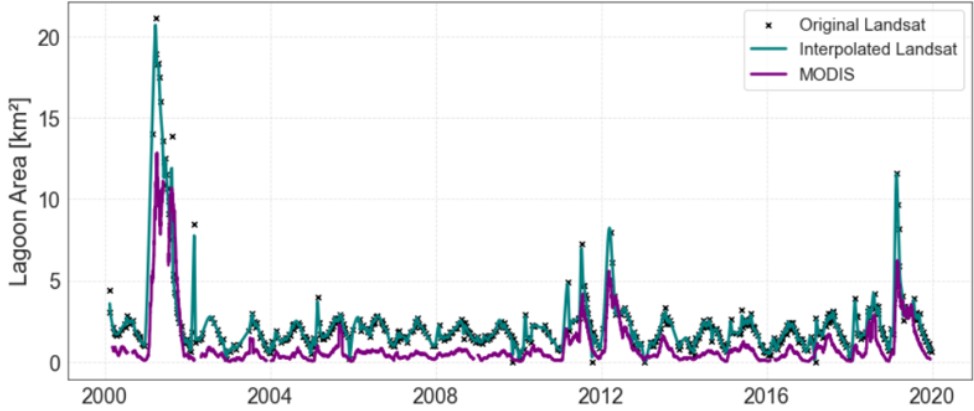

**Figure B6.** Time series of the shallow, salty lagoon area from Landsat, Interpolated Landsat and MODIS images.

*Data availability.* The downscaled datasets generated for each Hydrological Response Unit (HRU) defined in this study are openly available at: https://doi.org/10.17605/OSF.IO/6BG2W. The original datasets used as input for the downscaling are publicly available from their respective repositories.

*Author contributions.* **FAC:** Conceptualization, Methodology, Formal analysis and Discussion of results, Writing - original draft, Writing - review and editing. **OH:** Conceptualization, Methodology, Discussion of results, Writing - review and editing. **PBV:** Methodology, Discussion of results, Writing - review and editing. **FS:** Funding acquisition, Conceptualization, Methodology, Discussion of results, Writing - review and editing

*Competing interests.* The authors declare that they have no known competing financial interests or personal relationships that could have appeared to influence the work reported in this paper.

*Acknowledgements.* This research received financial support from the Agencia Nacional de Investigación y Desarrollo (ANID) through the projects ANID/ANILLO/ATE230006 and ANID/ FONDECYT/ 1251067. F. Aguirre-Correa also gratefully thanks ANID for providing financial support through the PhD scholarship BECAS/ DOCTORADO NACIONAL/ 21211730. The authors also acknowledge the research support provided by CEDEUS, ANID/ FONDAP/ 1523A0004. The authors finally thank Javiera Boada for the constructive discussions in the early stage of this research.

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
