# Peer review of "Questioning the Endorheic Paradigm: Water Balance dynamics in the Salar del Huasco basin, Chile"

_EGUsphere, 2025_

## Author Comment (AC1)

**Responses to Reviews of Egusphere-2025-2984: "Questioning the Endorheic Paradigm: Water Balance dynamics in the Salar del Huasco basin, Chile" By Francisca Aguirre-Correa et al.**

We sincerely thank Howard Wheater for taking the time to review our manuscript and for the constructive comments. For clarity, we have grouped some related comments by theme and provided joint responses where appropriate. We answered the comments in blue font.

This paper addresses a critically important issue for arid land hydrology and water management and provides an important case study. It is generally very well written and clearly explained. However, I have two important reservations.

The conclusions depend on satellite-derived forcing data applied to a hydrological model. The interpretation of missing water is wholly dependent on the validity of the forcing data, but there is limited discussion of the use of local data to improve these products, and crucially no attempt to quantify the potential errors and their impact on the water balance conclusions. Some effort to quantify the effects of errors on the water balance conclusions is essential.

Line 151: 'first order approximation'. I note no discussion as yet of the likely error bounds on the satellite estimates of precip and evaporation, but this is crucial for the data interpretation! The use of local data to improve the products is summarized rather briefly in Appendix B. More information would be helpful here, e.g. the local data available. The plots in App B do indicate quite large residual scatter. Some efforts to quantify likely errors and incorporate them in the analysis are in my view essential to the credibility of the conclusions.

We thank the reviewer for highlighting this essential point, which overlaps with a comment raised by Reviewer #2. Accordingly, we have provided a consistent response to ensure clarity and alignment.

We agree that the validity of our conclusions depends on the reliability of the satellite-derived datasets. To better explain the corrections of the satellite-derived data and quantify their uncertainty on the basin-scale water balance conclusions, we will make the following revisions in the manuscript:

1.  Expand description of local data and corrections: We will provide detailed metadata for the DGA precipitation and temperature stations used for bias correction, including their locations, elevations, record lengths, and error metrics of the corrected satellite-derived timeseries (PBIAS, RMSE, $R^2$). These will be presented in Tables B1 and B2 to complement Fig. B2 in Appendix B (see below), which will be referenced in the main text when introducing the satellite-derived datasets. In light of this comment, we would like to note that we identified a minor error in Fig. B2, specifically in the color legend. This will be corrected and updated in the revised manuscript (see Fig. R1).

[Figure]

Figure R1: Original (black dots) and corrected CR2MET data (blue dots for precipitation and red dots for air temperature) for the Salar del Huasco basin. A machine learning bias correction approach was implemented (Fig. B1).

**Table B1.** Summary of validation metrics for daily precipitation at DGA meteorological stations used for CR2MET bias correction. The colors in parentheses correspond to station markers in Fig. B2. PBIAS represents the percent bias computed for all days with recorded precipitation $> 1$ mm, RMSE is the root mean square error, and $R^2$ is the coefficient of determination. The basin median signed bias is derived from aggregating all station records into a single time series.

| Station | Lat/Lon | Elevation (m asl) | Record length | PBIAS (%) | RMSE (mm day$^{-1}$) | $R^2$ |
|---|---|---|---|---|---|---|
| Coyacagua (green) | -20.04°/-68.83° | 4013 | 14326 | 22.4 | 1.05 | 0.67 |
| Río Collacagua (red) | -20.10°/-68.84° | 3853 | 612 | 11.4 | 0.56 | 0.86 |
| Diablo Marca (orange) | -20.10°/-68.97° | 4585 | 1397 | 0.7 | 0.93 | 0.69 |
| Sillillica (purple) | -20.17°/-68.74° | 3840 | 1427 | -0.3 | 1.03 | 0.79 |
| Salar Huasco (black) | -20.28°/-68.89° | 3800 | 457 | 26.1 | 0.81 | 0.49 |
| Altos del Huasco (brown) | -20.32°/-68.89° | 4044 | 1449 | 2.6 | 0.62 | 0.76 |
| Basin median signed bias (%) | | -9.1 | | | | |

**Table B2.** Summary of validation metrics for daily air temperature at DGA meteorological stations used for CR2MET bias correction. The colors in parentheses correspond to station markers in Fig. B2. PBIAS represents the percent bias, RMSE is the root mean square error, and $R^2$ is the coefficient of determination. The basin median signed bias is derived from aggregating all station records into a single time series.

| Station | Lat/Lon | Elevation (m asl) | Record length | PBIAS (%) | RMSE (°C) | $R^2$ |
|---|---|---|---|---|---|---|
| Coyacagua (green) | -20.04°/-68.83° | 4013 | 11383 | -0.2 | 1.52 | 0.86 |
| Salar Huasco (black) | -20.28°/-68.89° | 3800 | 300 | 3.9 | 1.24 | 0.88 |
| Basin median signed bias (%) | | -0.9 | | | | |

2. Quantification of error ranges: We will quantify the uncertainties in the satellite-derived data and incorporate this analysis into the discussion, highlighting how these errors may influence our conclusions.

- Precipitation (P): Nearly all individual DGA stations show positive percent bias (PBIAS) values, indicating that bias-corrected CR2MET precipitation mainly overestimates the measurements at the station level, with magnitudes ranging from -0.3% to +26% (Table B1). The largest errors occur at lower-elevation stations, where total precipitation is low and satellite products tend to overestimate the few convective events that do occur. To evaluate the implications for the water balance at the basin scale, we aggregated all station records into a combined time series. This yielded a basin median signed bias of ~-10%, which we adopt as a representative error bound for precipitation (±10%). We chose the basin-wide median rather than the mean of station biases because the median provides a more robust estimate of central tendency, less influenced by extreme local values, and more consistent with the scale at which the water balance is assessed. However, note that our water balance results indicate evaporation exceeds precipitation, and since the precipitation product is more likely to be overestimated, as evidenced by the positive PBIAS (Table B1), the imbalance we report can be considered as conservative in terms of precipitation (true precipitation deficit is likely larger than our estimates suggest).

- Temperature (T): For air temperature, the available stations showed very small biases (<4%), with a basin-wide median signed bias of -0.9%. This indicates that after correction CR2MET temperature fields are in close agreement with in-situ observations, supporting their reliability as inputs to the rainfall-runoff model. This outcome is also consistent with the general expectation that satellite and reanalysis products reproduce temperature more accurately than precipitation.

- Evaporation (E): Direct local validation of evaporation was limited due to the absence of flux towers or lysimeter measurements in the basin. However, we will now compare EEFLUX-derived actual evaporation against daily evaporation estimated from pan measurements, adjusted using a pan coefficient of 0.7 as recommended by DGA (2009). These observations were collected at a representative site within the salt flat nucleus sub-basin (GP Consultores, 2008), the area with the highest evaporative fluxes in the study site. EEFLUX-derived actual evaporation underestimates the observed evaporation by approximately -18%, with a root mean square error (RMSE) of 1.41 mm day$^{-1}$ (Fig. R2). EEFLUX effectively reproduces the temporal variability captured by the tank observations, and its magnitude remains within a plausible range (Fig. R2). Furthermore, since EEFLUX/METRIC retrievals are strongly constrained by temperature, and satellite-derived temperature fields closely align with ground observations, confidence in the accuracy of the evaporation estimates is further supported. Nonetheless, because this comparison is limited to a single site and relies on pan measurements rather than actual evaporation, and considering that published evaluations of METRIC/EEFLUX in arid and mountainous regions have reported uncertainties of ~5-20% (Nisa et al., 2021; Wasti, 2020; Lima et al., 2020; Madugundu et al., 2017), we will adopt a conservative ±20% error bound for evaporation in our basin-scale water balance analysis. It is important to note, however, that the reported imbalance may be also conservative with respect to evaporation, which could be underestimated in this high-evaporation environment, as observed in Fig. R1.

[Figure]

**Figure R2:** Comparison of EEFLUX-derived daily evaporation at the salt nucleus sub-basin (green) with ground-based observations of potential evaporation (black) at a representative location in the salt flat nucleus sub-basin (GP Consultores, 2008). Both time series have been smoothed using a 10-day rolling average to highlight temporal variability.

3.  Analysis of implication of these errors into the basin water balance: Our baseline results indicate that basin-wide evaporation exceeds precipitation by a factor of ~1.5. To test whether plausible uncertainties could account for the imbalance, we will apply representative error bounds of ±10% for precipitation and ±20% for evaporation. Even under the most favorable scenario (increasing precipitation by 10% and decreasing evaporation by 20%), evaporation still exceeds precipitation and the imbalance persists. Based on our discussion, we hypothesize that the observed imbalance may reflect the influence of transboundary or interbasin groundwater flows not represented in the current hydrological model. We will now stress that future efforts should incorporate hydrogeological mapping and assessments geological connectivity, particularly toward neighboring Bolivian basins, to evaluate whether Salar del Huasco receives groundwater inflows from adjacent regions. This could help reconcile the apparent precipitation-evaporation deficit. Nevertheless, we will also emphasize in the revised manuscript the need for further validation of satellite-derived inputs against additional local observations across the basin, particularly for evaporation and air temperature for which only limited ground data are available, to strengthen the robustness of the water balance assessment.

Despite these limitations, we would like to highlight that the use of satellite-derived data represents a significant improvement in capturing the spatiotemporal variability of precipitation and evaporation in the basin. In the Altiplano region, convective storms and strong spatial heterogeneity limit the reliability of traditional approaches based on topographic gradients, as originally used in the rainfall-runoff model. While uncertainties in satellite products remain, our approach therefore provides a more realistic representation of hydrological dynamics in the Salar del Huasco, particularly of the interplay between precipitation, evaporation, and recharge. With this uncertainty analysis we will add transparency and robustness to our analysis, and we believe it will strengthen the overall contribution of the manuscript by providing a valuable first-order assessment of basin-scale water availability under the current climatic and data constraints.

The discussion of the seasonal dynamics of groundwater recharge depends on modelling results, but the model was calibrated using only surface flows. It seems that the groundwater dynamics are largely unconstrained. Some validation with local groundwater data would be invaluable. If not, at least there should be appropriate caveats.

Fig 3 results. The model that was used to simulate groundwater recharge was calibrated on observed river flows. So this provides only very limited information to define the dynamics of groundwater recharge fluxes. Were there no groundwater observations available to calibrate/validate this important component?

We acknowledge that the recharge component is less constrained than precipitation and evaporation because calibration relies solely on surface flow records. Local groundwater observations do exist, but they are sparse and spatially discontinuous, making them unsuitable for direct quantitative calibration or validation of groundwater recharge estimates. For this reason, we opted not to compare model outputs with these heterogeneous datasets. However, although not validated here against groundwater levels, the same model has been applied in the Salar del Huasco and other basins, where its recharge estimates were incorporated into groundwater flow models that performed well (Blin et al., 2022; Blin & Suárez, 2023). For instance, the Uribe et al. (2015) model was used by Blin et al. (2022) to estimate aquifer recharge in the basin, which was then further employed to drive the groundwater model that represented well the water levels observed throughout the basin. Therefore, although this model is calibrated with river discharge, it does provide a plausible spatiotemporal distribution of recharge that is consistent with observations from groundwater wells in the Salar de Huasco aquifer (see simulated and observed well levels in Blin et al., 2022). This gives confidence that our recharge outputs are reasonable as first-order estimates.

To strengthen the manuscript, we will:

1. Clarify the role of the model: Section 2.2 now will explicitly state that the model provides inferred recharge fluxes based on surface-water calibration, rather than direct simulations of groundwater levels. We will also emphasize that the model is primarily used to represent the partitioning and timing of surface runoff and recharge.

2. Note previous successful applications: We will now highlight in the manuscript that the same model structure has been successfully used in the Salar del Huasco and other basins, where its recharge estimates were incorporated into groundwater models (e.g., MODFLOW in Blin et al., 2022), where observed water levels are well represented.

3. Added current limitations and future directions: We will now reinforce the ideas that (i) recharge estimates should be viewed as first-order approximations; and (ii) future work should integrate groundwater-level or tracer data where available to validate recharge dynamics.

Line 13: replace 'insinuate' by 'imply'

We will update the manuscript accordingly.

Line 37: composed of

We will update the manuscript accordingly.

Line 91 of the Uribe et al…

Line 112 the Uribe….

Line 131 the Uribe…

Line 156 the Uribe… please correct throughout – including Fig 3 caption

We will add "the" before "Uribe et al." throughout the manuscript, including Figure 3 caption.

Line 94: into

We will update the manuscript accordingly.

Section 2.2 para 2 specify the model and forcing data time steps. I assume daily??

We will clarify that model and forcing data time steps are daily.

Line 116: data … are used…

We will update the manuscript accordingly.

Line 118: please clarify what is meant by evaporation here. It isn't obvious until line 137.

We will clarify the definition in the manuscript: *"In this study, evaporation refers specifically to satellite-retrieved actual evaporation from EEFLUX/METRIC, rather than potential evaporation or model-estimated fluxes."*

References

Blin, N., Hausner, M., Leray, S., Lowry, C., Suárez, F. (2022). Potential impacts of climate change on an aquifer in the arid altiplano, northern chile: The case of the protected wetlands of the salar del huasco basin. Journal of Hydrology: Regional Studies 39, 100996. https://doi.org/10.1016/j.ejrh.2022.100996

Blin, N., & Suárez, F.. (2023). Evaluating the contribution of satellite-derived evapotranspiration in the calibration of numerical groundwater models in remote zones using the EEFlux tool, Science of The Total Environment, Volume 858, Part 1, 159764, ISSN 0048-9697, https://doi.org/10.1016/j.scitotenv.2022.159764.

Collahuasi - GP Consultores. (2008). Información sobre evaporación provista por Compañía Minera Doña Inés de Collahuasi. Informes inéditos.

DGA. (2009). Sistema Piloto I Región: Salar del Huasco. In Levantamieno hidrogeológico para el desarrollo de nuevas fuentes de agua en áreas prioritarias de la zona norte de Chile, Regiones XV, I, II y III.

Madugundu, R., Al-Gaadi, K. A., Tola, E., Hassaballa, A. A., and Patil, V. C. (2017). Performance of the METRIC model in estimating evapotranspiration fluxes over an irrigated field in Saudi Arabia using Landsat-8 images, Hydrol. Earth Syst. Sci., 21, 6135–6151, https://doi.org/10.5194/hess-21-6135-2017.

Nisa, Z., Khan, M. S., Govind, A., Marchetti, M., Lasserre, B., Magliulo, E., & Manco, A. (2021). Evaluation of SEBS, METRIC-EEFlux, and QWaterModel Actual Evapotranspiration for a Mediterranean Cropping System in Southern Italy. *Agronomy*, *11*(2), 345. https://doi.org/10.3390/agronomy11020345

Lima, João & Sánchez-Tomás, Juan & G. Piqueras, José & Sobrinho, José & C. Viana, Paula & S. Alves, Aline. (2020). Evapotranspiration of sorghum from the energy balance by METRIC and STSEB. Revista Brasileira de Engenharia Agrícola e Ambiental. 24. 24-30. 10.1590/1807-1929/agriambi.v24n1p24-30.

Wasti, Shailaja. (2020). Estimation of land surface Evapotranspiration in Nepal using Landsat based METRIC model. 10.48550/arXiv.2007.13922.

---

## Author Comment (AC2)

**Responses to Reviews of Egusphere-2025-2984: "Questioning the Endorheic Paradigm: Water Balance dynamics in the Salar del Huasco basin, Chile" By Francisca Aguirre-Correa et al.**

We thank the Reviewer for spending time revising our work and for the constructive comments. For clarity, we have grouped some related comments by theme and provided joint responses where appropriate. We answered the comments in blue font.

This manuscript presents an investigation of the spatiotemporal variability of groundwater recharge and upwelling in the Salar del Huasco basin, focusing on its interactions with precipitation, evaporation, and overall water balance dynamics in this arid endorheic system located in the Chilean Altiplano. The study addresses an important and timely topic with relevance to hydrological processes in data-scarce, high-altitude regions and provides valuable insights into the interactions among precipitation, evaporation, and groundwater recharge under conditions of extreme water scarcity. The manuscript is well structured, and the results are presented clearly. The work has the potential to make a contribution to the understanding of water balance processes in arid endorheic systems. However, certain aspects would benefit from clarification and further elaboration.

Some specific comments are provided below for consideration:

Lines 9-10: The message of this sentence is not entirely clear and would benefit from rephrasing to improve coherence and accuracy. Groundwater itself cannot "lead to recharge," since recharge is the process that feeds or replenishes groundwater storage. The intended meaning seems to be that groundwater storage decreases, or that recharge reaches a minimum following the dry period.

We agree and we will rephrase the sentence for accuracy. The intended meaning is that following the dry period, groundwater sustains high evaporation rates, while recharge declines to a minimum: *"Our findings highlight that when summer rainfall ceases, groundwater becomes the main water source supporting high evaporation rates, and recharge reaches a minimum by the end of autumn that persists until the end of the year."*

Line 11: The term *"competition"* is not ideal in this hydrological context. Please replace *"trade-off" or similar words* to describe the contrasting relationship between groundwater recharge and evaporation.

We agree and we will now use the term "trade-off" throughout the manuscript, which better reflects the contrasting relationship between recharge and evaporation.

Lines 84-87 It would be helpful to express the total annual discharge in units of mm yr$^{-1}$, in addition to the volumetric flow rates (m³ s$^{-1}$), to facilitate direct comparison with precipitation and evaporation values reported elsewhere in mm yr$^{-1}$ in the manuscript.

While we report spring flows in m³s$^{-1}$, conversion to mm yr$^{-1}$ requires knowledge of the contributing recharge area, which is not available for these springs. To at least provide an order of magnitude, we will use the salt-flat nucleus sub-basin area (48.22 km²) as a reference. Under this assumption, the total spring discharge of $\sim$0.2 m³ s$^{-1}$ corresponds to roughly 130 mm yr$^{-1}$ (order of magnitude of $\sim$10$^2$ mm yr$^{-1}$).

We will now report this order-of-magnitude estimate in the manuscript, while we will continue to present the observed values in m³s⁻¹ that can be directly compared to precipitation and evaporation fluxes also expressed in m³/s in Figs. 6 and 7.

Lines 131-132: The authors state that the Uribe et al. (2015) rainfall-runoff model is driven solely by precipitation, and in the model setup, evaporation occurs only in response to rainfall events. Please clarify the rationale for selecting this model despite this limitation, and discuss how this assumption may affect the results and the interpretation of the water balance.

We thank the reviewer for this comment. We used the Uribe et al. (2015) rainfall-runoff model because it was originally developed for the Salar de Huasco and is based on hydrological response units (HRUs) already defined for this basin and widely used in Altiplano studies (e.g., Acosta, 2004; Blin et al., 2022; Yañez-Morroni et al., 2024a,b). This ensures consistency with earlier work and takes advantage of a model structure already adapted to the physical, geological, and climatic conditions of the catchment.

While the original model formulation represents evaporation mainly as an event-driven process tied to rainfall, we adapted the setup to better capture off-season evaporation by introducing an evaporation component that operates independently of rainfall inputs. This allows the model to account for evaporation during the long dry season, which is particularly important in high-altitude basins such as the Salar de Huasco. In our study, the model is therefore used primarily to represent the partitioning and timing of surface runoff and groundwater recharge, while basin-scale closure is derived independently from bias corrected satellite-based evaporation and precipitation considering that no changes in groundwater levels have been observed in the basin (see Blin et al., 2022). However, we acknowledge that the modified model may misrepresent water losses, particularly during the dry season, due to uncertainties in satellite-derived evaporation data, which could in turn impact the accuracy of groundwater recharge estimates.

We will now explain in the manuscript the rationale for selecting this model and we will highlight that the same model structure has been successfully used in the Salar del Huasco and other basins. We will also discuss the limitation that it may affect recharge estimates, and we will emphasize that the model is primarily used to represent the partitioning and timing of surface runoff and recharge.

Figure 3: It is unclear how "monthly mean variability" can be represented in $mm\ yr^{-1}$, since those units denote annual totals or rates. It appears that Figure 3 presents monthly mean values on the x-axis; therefore, the corresponding y-axis units should likely be expressed in $mm$ (or $mm\ month^{-1}$), rather than $mm\ yr^{-1}$. Please clarify the unit definition. If the intention is to show monthly distributions, the values should be expressed in $mm\ month^{-1}$ (or simply $mm$), or the caption should explicitly state that the data have been annualized (e.g., monthly averages scaled to their annual equivalents). Please clarify the units and calculation method to ensure the figure is interpreted correctly. Otherwise, the comparison between rainfall and evaporation may be misleading, as expressing monthly values in $mm\ yr^{-1}$ artificially inflates their magnitude by a factor of 12.

The results described between lines 207–225 also seem to be based on monthly values presented in Figure 3. If this is the case, the current labeling in $mm\ yr^{-1}$ is inconsistent and may cause confusion. Conversely, if the data were converted to annual equivalents, it would be preferable to present Figure 3 using an annual scale rather than monthly intervals, to maintain conceptual consistency.

A similar applies to Figure 4, particularly the right panels (a–c). The values are presented as a function of months, suggesting that the data represent *monthly means*. If the intention is to illustrate the mean variation across available months (e.g., showing typical January, February, etc., values), then the units should

correspond to the monthly time step. Alternatively, if the values have been annualized, please clarify this in the caption and consider adjusting the x-axis to represent annual rather than monthly time intervals for conceptual consistency.

*The reviewer is correct that Figures 3 and 4, and also Figure 5, show monthly means. All water balance components in the manuscript (precipitation, evaporation, and recharge) are presented as monthly averages but expressed in annual-equivalent units (mm yr⁻¹). This rescaling was applied to keep a consistent vertical scale across fluxes and to avoid very small values in mm month⁻¹, particularly for recharge, which is our main focus. The x-axis remains monthly to highlight seasonal variability, while the y-axis in mm yr⁻¹ ensures comparability across figures. The captions will now make this explicit:*

*For Fig. 3: "Values are presented for the 1980 - 2019 period and represent monthly means, rescaled to annual equivalents (mm year-1) for consistency across all water balance components."*

*For Figs. 4 and 5: "Values represent monthly means, rescaled to annual equivalents (mm year-1) for consistency across all water balance components."*

*We will also clarify this in the text when necessary: "Groundwater recharge is displayed as monthly averages scaled to their annual equivalents in mm year⁻¹".*

Line 196 -206: In this section, the results are presented without a corresponding reference to any figure or table. Please specify which figure (or sub-figure) illustrates these findings so that readers can easily locate and interpret the results. Clear cross-referencing between text and figures would greatly improve readability and traceability of the analysis for the Results and Discussion section

*We will add explicit cross-references to Fig. 3 where the results are discussed, and we will also carefully review the manuscript to ensure that all figure references are consistent.*

Line 244: The text refers to "S1-type HRUs with an annual mean of ∼600 mm yr⁻¹ (see Fig. 2)," but Figure 2 does not display any numerical values or spatial distribution of annual means; it only shows the HRU classification. Please clarify whether these mean values are derived from another figure, dataset, or analysis step, and adjust the figure reference accordingly.

*We thank the reviewer for this correction. We will update this reference to Fig. 4b.*

The Discussion section would benefit from a more detailed consideration of the limitations associated with the satellite-derived datasets and the assumptions of the applied model. Currently, these aspects are only briefly mentioned in lines 383–386. Given that the presented data are derived rather than directly measured, it is important to discuss the potential uncertainties arising from data correction methods, model parameterization, and structural assumptions.

*We thank Reviewer #2 for this comment, which overlaps with a point raised by Reviewer #1. Accordingly, we have provided a consistent response to ensure clarity and alignment. Overall, we agree that the validity of our conclusions depends on the reliability of the satellite-derived datasets and the underlying model assumptions. We acknowledge that the manuscript will benefit from a more thorough and transparent discussion of these limitations, which we will now expand accordingly.*

To better explain the corrections of the satellite-derived data and quantify their uncertainty on the basin-scale water balance conclusions, we will make the following revisions in the manuscript:

1. Expand description of local data and corrections: We will provide detailed metadata for the DGA precipitation and temperature stations used for bias correction, including their locations, elevations, record lengths, and error metrics of the corrected satellite-derived timeseries (PBIAS, RMSE, $R^2$). These will be presented in Tables B1 and B2 to complement Fig. B2 in Appendix B (see below), which will be referenced in the main text when introducing the satellite-derived datasets. In light of this comment, we would like to note that we identified a minor error in Fig. B2, specifically in the color legend. This will be corrected and updated in the revised manuscript (see Fig. R1).

[Figure]

Figure R1: Original (black dots) and corrected CR2MET data (blue dots for precipitation and red dots for air temperature) for the Salar del Huasco basin. A machine learning bias correction approach was implemented (Fig. B1).

**Table B1.** Summary of validation metrics for daily precipitation at DGA meteorological stations used for CR2MET bias correction. The colors in parentheses correspond to station markers in Fig. B2. PBIAS represents the percent bias computed for all days with recorded precipitation > 1 mm, RMSE is the root mean square error, and $R^2$ is the coefficient of determination. The basin median signed bias is derived from aggregating all station records into a single time series.

| Station | Lat/Lon | Elevation (m asl) | Record length | PBIAS (%) | RMSE (mm day$^{-1}$) | $R^2$ |
|---|---|---|---|---|---|---|
| Coyacagua (green) | -20.04°/-68.83° | 4013 | 14326 | 22.4 | 1.05 | 0.67 |
| Río Collacagua (red) | -20.10°/-68.84° | 3853 | 612 | 11.4 | 0.56 | 0.86 |
| Diablo Marca (orange) | -20.10°/-68.97° | 4585 | 1397 | 0.7 | 0.93 | 0.69 |
| Sillillica (purple) | -20.17°/-68.74° | 3840 | 1427 | -0.3 | 1.03 | 0.79 |
| Salar Huasco (black) | -20.28°/-68.89° | 3800 | 457 | 26.1 | 0.81 | 0.49 |
| Altos del Huasco (brown) | -20.32°/-68.89° | 4044 | 1449 | 2.6 | 0.62 | 0.76 |
| Basin median signed bias (%) | -9.1 | | | | | |

**Table B2.** Summary of validation metrics for daily air temperature at DGA meteorological stations used for CR2MET bias correction. The colors in parentheses correspond to station markers in Fig. B2. PBIAS represents the percent bias, RMSE is the root mean square error, and $R^2$ is the coefficient of determination. The basin median signed bias is derived from aggregating all station records into a single time series.

| Station | Lat/Lon | Elevation (m asl) | Record length | PBIAS (%) | RMSE (°C) | $R^2$ |
|---|---|---|---|---|---|---|
| Coyacagua (green) | -20.04°/-68.83° | 4013 | 11383 | -0.2 | 1.52 | 0.86 |
| Salar Huasco (black) | -20.28°/-68.89° | 3800 | 300 | 3.9 | 1.24 | 0.88 |
| Basin median signed bias (%) | -0.9 | | | | | |

2. Quantification of error ranges: We will quantify the uncertainties in the satellite-derived data and incorporate this analysis into the discussion, highlighting how these errors may influence our conclusions.

- Precipitation (P): Nearly all individual DGA stations show positive percent bias (PBIAS) values, indicating that bias-corrected CR2MET precipitation mainly overestimates the measurements at the station level, with magnitudes ranging from -0.3% to +26% (Table B1). The largest errors occur at lower-elevation stations, where total precipitation is low and satellite products tend to overestimate the few convective events that do occur. To evaluate the implications for the water balance at the basin scale, we aggregated all station records into a combined time series. This yielded a basin median signed bias of ~-10%, which we adopt as a representative error bound for precipitation (±10%). We chose the basin-wide median rather than the mean of station biases because the median provides a more robust estimate of central tendency, less influenced by extreme local values, and more consistent with the scale at which the water balance is assessed. However, note that our water balance results indicate evaporation exceeds precipitation, and since the precipitation product is more likely to be overestimated, as evidenced by the positive PBIAS (Table B1), the imbalance we report can be considered as conservative in terms of precipitation (true precipitation deficit is likely larger than our estimates suggest).

- Temperature (T): For air temperature, the available stations showed very small biases (<4%), with a basin-wide median signed bias of -0.9%. This indicates that after correction CR2MET temperature fields are in close agreement with in-situ observations, supporting their reliability as inputs to the rainfall-runoff model. This outcome is also consistent with the general expectation that satellite and reanalysis products reproduce temperature more accurately than precipitation.

- Evaporation (E): Direct local validation of evaporation was limited due to the absence of flux towers or lysimeter measurements in the basin. However, we will now compare EEFLUX-derived actual evaporation against daily evaporation estimated from pan measurements, adjusted using a pan coefficient of 0.7 as recommended by DGA (2009). These observations were collected at a representative site within the salt flat nucleus sub-basin (GP Consultores, 2008), the area with the highest evaporative fluxes in the study site. EEFLUX-derived actual evaporation underestimates the observed evaporation by approximately -18%, with a root mean square error (RMSE) of 1.41 mm day$^{-1}$ (Fig. R2). EEFLUX effectively reproduces the temporal variability captured by the tank observations, and its magnitude remains within a plausible range (Fig. R2). Furthermore, since EEFLUX/METRIC retrievals are strongly constrained by temperature, and satellite-derived temperature fields closely align with ground observations, confidence in the accuracy of the evaporation estimates is further supported. Nonetheless, because this comparison is limited to a single site and relies on pan measurements rather than actual evaporation, and considering that published evaluations of METRIC/EEFLUX in arid and mountainous regions have reported uncertainties of ~5-20% (Nisa et al., 2021; Wasti, 2020; Lima et al., 2020; Madugundu et al., 2017), we will adopt a conservative ±20% error bound for evaporation in our basin-scale water balance analysis. It is important to note, however, that the reported imbalance may be also conservative with respect to evaporation, which could be underestimated in this high-evaporation environment, as observed in Fig. R1.

[Figure]

**Figure R2:** Comparison of EEFLUX-derived daily evaporation at the salt nucleus sub-basin (green) with ground-based observations of potential evaporation (black) at a representative location in the salt flat nucleus sub-basin (GP Consultores, 2008). Both time series have been smoothed using a 10-day rolling average to highlight temporal variability.

3. Analysis of implication of these errors into the basin water balance: Our baseline results indicate that basin-wide evaporation exceeds precipitation by a factor of ~1.5. To test whether plausible uncertainties could account for the imbalance, we will apply representative error bounds of ±10% for precipitation and ±20% for evaporation. Even under the most favorable scenario (increasing precipitation by 10% and decreasing evaporation by 20%), evaporation still exceeds precipitation and the imbalance persists. Based on our discussion, we hypothesize that the observed imbalance may reflect the influence of transboundary or interbasin groundwater flows not represented in the

current hydrological model. We will now stress that future efforts should incorporate hydrogeological mapping and assessments of geological connectivity, particularly toward neighboring Bolivian basins, to evaluate whether Salar del Huasco receives groundwater inflows from adjacent regions. This could help reconcile the apparent precipitation-evaporation deficit. Nevertheless, we will also emphasize in the revised manuscript the need for further validation of satellite-derived inputs against additional local observations across the basin, particularly for evaporation and air temperature for which only limited ground data are available, to strengthen the robustness of the water balance assessment.

Despite these limitations, we would like to highlight that the use of satellite-derived data represents a significant improvement in capturing the spatiotemporal variability of precipitation and evaporation in the basin. In the Altiplano region, convective storms and strong spatial heterogeneity limit the reliability of traditional approaches based on topographic gradients, as originally used in the rainfall-runoff model. While uncertainties in satellite products remain, our approach therefore provides a more realistic representation of hydrological dynamics in the Salar del Huasco, particularly of the interplay between precipitation, evaporation, and recharge. With this uncertainty analysis we will add transparency and robustness to our analysis, and we believe it will strengthen the overall contribution of the manuscript by providing a valuable first-order assessment of basin-scale water availability under the current climatic and data constraints.

We will also acknowledge that our rainfall-runoff modeling framework provides a first-order approximation of groundwater recharge and is subject to sources of uncertainty. To improve transparency, we will expand the discussion in both Appendix A and the main text to clarify the key assumptions and their potential impact on model results:

1. Satellite-derived input limitations: Despite bias corrections based on ground observations, residual errors remain in satellite-derived precipitation, temperature, and evaporation fields, particularly in sparsely monitored areas. These uncertainties propagate through the model and can lead to misrepresentation of water inputs and outputs, thereby influencing the estimated recharge and balance outcomes.

2. Dry-season evaporation parameterization: Evaporation during the dry season is not dynamically simulated but instead prescribed using satellite-derived input. This may lead to under- or overestimation of seasonal water losses, which can directly affect recharge estimates, especially during the dry season.

3. Omission of cryospheric and freeze-thaw processes: The model does not account for snow accumulation, snowmelt, or soil freeze-thaw processes which, although not dominant, have been observed in the Salar del Huasco basin. Excluding these processes may lead to inaccurate timing and magnitude of infiltration and recharge, especially during transitional seasons (e.g., spring melt), potentially biasing recharge estimates.

4. Streamflow data limitations: The discharge data used for model calibration are subject to observational uncertainty, especially during low or ephemeral flow conditions. These limitations can reduce the reliability of the calibration and impact the model's ability to correctly represent water partitioning processes, including infiltration and runoff.

5.  Model parameter uncertainty: The model includes 15 calibrated parameters, increasing the risk of equifinality, where multiple parameter sets yield similar performance but differ in internal process representation. This compromises the uniqueness and robustness of the estimated recharge. Following recent recommendations (e.g., Yáñez-Morroni et al., 2024a,b), we will recommend that future work simplify the model by reducing the number of parameters and using less complex formulations, to improve the clarity and reliability of parameter estimates.

We would like to emphasize that, despite the limitations in the rainfall-runoff model, it offers a spatially distributed and process-oriented framework for analyzing groundwater recharge, and it enables exploration of the dominant hydrological controls in the Salar del Huasco basin. Nevertheless, we will emphasize that the results should be interpreted as indicative, and future refinement will benefit from improved data coverage and model structure.

**References**

Acosta, O. (2004). Impacto de las extracciones de agua subterránea en el salar del huasco. Master's thesis, Tesis de Máster, Universidad Politécnica de Cataluña. Barcelona, España.

Blin, N., Hausner, M., Leray, S., Lowry, C., Suárez, F. (2022). Potential impacts of climate change on an aquifer in the arid altiplano, northern chile: The case of the protected wetlands of the salar del huasco basin. Journal of Hydrology: Regional Studies 39, 100996. https://doi.org/10.1016/j.ejrh.2022.100996

Blin, N., & Suárez, F.. (2023). Evaluating the contribution of satellite-derived evapotranspiration in the calibration of numerical groundwater models in remote zones using the EEFlux tool, Science of The Total Environment, Volume 858, Part 1, 159764, ISSN 0048-9697, https://doi.org/10.1016/j.scitotenv.2022.159764.

Collahuasi - GP Consultores. (2008). Información sobre evaporación provista por Compañía Minera Doña Inés de Collahuasi. Informes inéditos.

DGA. (2009). Sistema Piloto I Región: Salar del Huasco. In Levantamieno hidrogeológico para el desarrollo de nuevas fuentes de agua en áreas prioritarias de la zona norte de Chile, Regiones XV, I, II y III.

Madugundu, R., Al-Gaadi, K. A., Tola, E., Hassaballa, A. A., and Patil, V. C. (2017). Performance of the METRIC model in estimating evapotranspiration fluxes over an irrigated field in Saudi Arabia using Landsat-8 images, Hydrol. Earth Syst. Sci., 21, 6135–6151, https://doi.org/10.5194/hess-21-6135-2017.

Nisa, Z., Khan, M. S., Govind, A., Marchetti, M., Lasserre, B., Magliulo, E., & Manco, A. (2021). Evaluation of SEBS, METRIC-EEFlux, and QWaterModel Actual Evapotranspiration for a Mediterranean Cropping System in Southern Italy. *Agronomy*, *11*(2), 345. https://doi.org/10.3390/agronomy11020345

Lima, João & Sánchez-Tomás, Juan & G. Piqueras, José & Sobrinho, José & C. Viana, Paula & S. Alves, Aline. (2020). Evapotranspiration of sorghum from the energy balance by METRIC and STSEB. Revista Brasileira de Engenharia Agrícola e Ambiental. 24. 24-30. 10.1590/1807-1929/agriambi.v24n1p24-30.

Wasti, Shailaja. (2020). Estimation of land surface Evapotranspiration in Nepal using Landsat based METRIC model. 10.48550/arXiv.2007.13922.

Yáñez-Morroni, G., Suárez, F., Muñoz, J.F., Lagos, M.S. (2024a). Hydrological modeling of the silala river basin. 1. model development and long-term groundwater recharge assessment. WIREs Water 11(1), 1690 https://doi.org/10.1002/wat2.1690

Yáñez-Morroni, G., Suárez, F., Muñoz, J.F., Lagos, M.S. (2024b). Hydrological modeling of the silala river basin. 2. validation of hydrological fluxes with contemporary data. WIREs Water 11(1), 1696 https://doi.org/10.1002/wat2.1696

---

## Author Response (AR1)

**Responses to Reviews of Egusphere-2025-2984: "Questioning the Endorheic Paradigm: Water Balance dynamics in the Salar del Huasco basin, Chile" By Francisca Aguirre-Correa et al.**

We thank the Reviewers for spending time revising our work. The Reviewers share main and specific concerns that we have grouped by theme and answered jointly. A few remaining, minor comments not shared between the reviewers, we addressed separately. We answered the comments in blue font, specifying how and where we modified the attached revised manuscript. We also attach a second version of our manuscript in which all the changes we made are highlighted.

Satellite-derived data uncertainty

*Reviewer 1.* The conclusions depend on satellite-derived forcing data applied to a hydrological model. The interpretation of missing water is wholly dependent on the validity of the forcing data, but there is limited discussion of the use of local data to improve these products, and crucially no attempt to quantify the potential errors and their impact on the water balance conclusions. Some effort to quantify the effects of errors on the water balance conclusions is essential.

*Reviewer 1.* Line 151: 'first order approximation'. I note no discussion as yet of the likely error bounds on the satellite estimates of precip and evaporation, but this is crucial for the data interpretation! The use of local data to improve the products is summarized rather briefly in Appendix B. More information would be helpful here, e.g. the local data available. The plots in App B do indicate quite large residual scatter. Some efforts to quantify likely errors and incorporate them in the analysis are in my view essential to the credibility of the conclusions.

*Reviewer 2.* The Discussion section would benefit from a more detailed consideration of the limitations associated with the satellite-derived datasets (…). Currently, these aspects are only briefly mentioned in lines 383–386. Given that the presented data are derived rather than directly measured, it is important to discuss the potential uncertainties arising from data correction methods (…).

We thank the Reviewers for highlighting this essential point, as we agree that the validity of our conclusions depends on the reliability of the satellite-derived datasets. To better explain the corrections of the satellite-derived data and quantify their uncertainty on the basin-scale water balance conclusions, we have made the following revisions to the manuscript:

1. Expanded description of local data and corrections: We now provide detailed metadata for the DGA precipitation and temperature stations used for bias correction, including their locations, elevations, record lengths, and error metrics of the corrected satellite-derived timeseries (PBIAS, RMSE, $R^2$). These are presented in Tables B1 and B2 in Appendix B (see below), which are now referenced in the main text when introducing the satellite-derived datasets. In light of this comment, we would like to note that we identified a minor error in Fig. B2, specifically in the color legend. This was corrected and updated in the revised manuscript (see Fig. R1).

[Figure]

Figure R1: Original (black dots) and corrected CR2MET data (blue dots for precipitation and red dots for air temperature) at DGA stations across the Salar del Huasco basin (see Tables B1 and B2). A machine learning bias correction approach was implemented (Fig. B1).

**Table B1.** Summary of validation metrics for daily precipitation at DGA meteorological stations used for CR2MET bias correction. The colors in parentheses correspond to station markers in Fig. B2. PBIAS represents the percent bias computed for all days with recorded precipitation > 1 mm, RMSE is the root mean square error, and $R^2$ is the coefficient of determination. The basin median signed bias is derived from aggregating all station records into a single time series.

| Station | Lat/Lon | Elevation (m asl) | Record length | PBIAS (%) | RMSE (mm day$^{-1}$) | $R^2$ |
|---|---|---|---|---|---|---|
| Coyacagua (green) | -20.04°/-68.83° | 4013 | 14326 | 22.4 | 1.05 | 0.67 |
| Río Collacagua (red) | -20.10°/-68.84° | 3853 | 612 | 11.4 | 0.56 | 0.86 |
| Diablo Marca (orange) | -20.10°/-68.97° | 4585 | 1397 | 0.7 | 0.93 | 0.69 |
| Sillillica (purple) | -20.17°/-68.74° | 3840 | 1427 | -0.3 | 1.03 | 0.79 |
| Salar Huasco (black) | -20.28°/-68.89° | 3800 | 457 | 26.1 | 0.81 | 0.49 |
| Altos del Huasco (brown) | -20.32°/-68.89° | 4044 | 1449 | 2.6 | 0.62 | 0.76 |
| Basin median signed bias (%) | | -9.1 | | | | |

**Table B2.** Summary of validation metrics for daily air temperature at DGA meteorological stations used for CR2MET bias correction. The colors in parentheses correspond to station markers in Fig. B2. PBIAS represents the percent bias, RMSE is the root mean square error, and $R^2$ is the coefficient of determination. The basin median signed bias is derived from aggregating all station records into a single time series.

| Station | Lat/Lon | Elevation (m asl) | Record length | PBIAS (%) | RMSE (°C) | $R^2$ |
|---|---|---|---|---|---|---|
| Coyacagua (green) | -20.04°/-68.83° | 4013 | 11383 | -0.2 | 1.52 | 0.86 |
| Salar Huasco (black) | -20.28°/-68.89° | 3800 | 300 | 3.9 | 1.24 | 0.88 |
| Basin median signed bias (%) | | -0.9 | | | | |

2. Quantification of error ranges: We quantified the uncertainties in the satellite-derived data and incorporated this analysis into the discussion, highlighting how these errors may influence our conclusions.

- Precipitation (P): Nearly all individual DGA stations show positive percent bias (PBIAS) values, indicating that bias-corrected CR2MET precipitation mainly overestimates the measurements at the station level, with magnitudes ranging from -0.3% to +26% (Table B1). The largest errors occur at lower-elevation stations, where total precipitation is low and satellite products tend to overestimate the few convective events that do occur. To evaluate the implications for the water balance at the basin scale, we aggregated all station records into a combined time series. This yielded a basin median signed bias of ∼-10%, which we adopted as a representative error bound for precipitation (±10%). We chose the basin-wide median rather than the mean of station biases because the median provides a more robust estimate of central tendency, less influenced by extreme local values, and more consistent with the scale at which the water balance is assessed. However, note that our water balance results indicate evaporation exceeds precipitation, and since the precipitation product is more likely to be overestimated, as evidenced by the positive PBIAS (Table B1), the imbalance we report can be considered as conservative in terms of precipitation (true precipitation deficit is likely larger than our estimates suggest).

- Temperature (T): For air temperature, the available stations show very small biases (<4%), with a basin-wide median signed bias of -0.9% (Table B2). This indicates that after correction CR2MET temperature fields are in close agreement with in-situ observations, supporting their reliability as inputs to the rainfall-runoff model. This outcome is also consistent with the general expectation that satellite and reanalysis products reproduce temperature more accurately than precipitation.

- Evaporation (E): Direct local validation of evaporation was limited due to the absence of long-term flux towers or lysimeter measurements in the basin. However, we now compare EEFLUX-derived actual evaporation against daily evaporation estimated from pan measurements, adjusted using a pan coefficient of 0.7 as recommended by DGA (2009). These observations were collected at a representative site within the salt flat nucleus sub-basin (GP Consultores, 2008), the area with the highest evaporative fluxes in the study site. EEFLUX-derived actual evaporation underestimates the observed evaporation by approximately -18%, with a root mean square error (RMSE) of 1.41 mm day$^{-1}$ (Fig. R2). EEFLUX effectively reproduces the temporal variability captured by the tank observations, and its magnitude remains within a plausible range (Fig. R2). Furthermore, since EEFLUX/METRIC retrievals are strongly constrained by temperature, and satellite-derived temperature fields closely align with ground observations, confidence in the accuracy of the evaporation estimates is further supported. Nonetheless, because this comparison is limited to a single site and relies on pan measurements rather than actual evaporation, and considering that published evaluations of METRIC/EEFLUX in arid and mountainous regions have reported uncertainties of ~5-20% (Nisa et al., 2021; Wasti, 2020; Lima et al., 2020; Madugundu et al., 2017), we adopted a conservative ±20% error bound for evaporation in our basin-scale water balance analysis.

[Figure]

**Figure R2:** Comparison of EEFLUX-derived daily evaporation at the salt nucleus sub-basin (green) with ground-based observations of potential evaporation (black) at a representative location in the salt flat nucleus sub-basin (GP Consultores, 2008). Both time series have been smoothed using a 10-day rolling average to highlight temporal variability.

3. Analysis of implication of these errors into the basin water balance: Our baseline results indicate that basin-wide evaporation exceeds precipitation by a factor of ~1.5. To test whether plausible uncertainties could account for the imbalance, we considered representative error bounds of ±10% for precipitation and ±20% for evaporation. Even under the most favorable scenario (increasing precipitation by 10% and decreasing evaporation by 20%), evaporation still exceeds precipitation and the imbalance persists. We now emphasize that plausible satellite data errors cannot explain the observed water imbalance. Instead, based on our discussion, we hypothesize that the observed imbalance may reflect the influence of transboundary or interbasin groundwater flows not represented in the current hydrological model. We now stress that future efforts should incorporate hydrogeological mapping and assessments of geological connectivity, particularly toward neighboring Bolivian basins, to evaluate whether Salar del Huasco receives groundwater inflows from adjacent regions. This could help reconcile the apparent precipitation-evaporation deficit. Nevertheless, we now also emphasize in the revised manuscript the need for further validation of satellite-derived inputs against additional local observations across the basin, particularly for evaporation and air temperature for which only limited ground data are available, to strengthen the robustness of the water balance assessment.

Despite these limitations, we would like to highlight that the use of satellite-derived data represents a significant improvement in capturing the spatiotemporal variability of precipitation and evaporation in the basin. In the Altiplano region, localized convective storms and strong spatial heterogeneity limit the reliability of traditional approaches based on topographic gradients, as originally used in the rainfall-runoff model. While uncertainties in satellite products remain, our approach therefore provides a more realistic representation of hydrological dynamics in the Salar del Huasco, particularly of the interplay between precipitation, evaporation, and recharge. With this uncertainty analysis we believe we added transparency and robustness to our analysis, and it strengthens the overall contribution of the manuscript by providing a valuable first-order assessment of basin-scale water availability under the current climatic and data constraints.

We have modified the revised manuscript as follows:

Lines 17-18: *"Also, validating satellite-derived inputs against additional local observations is essential to strengthen the reliability of the water balance assessment."*

Lines 128-133: *"Since the rainfall-runoff model results depend directly on the quality of the forcing data, we applied bias corrections using local ground-based observations to improve the reliability of the satellite products. A machine learning-based bias correction was developed to downscale CR2MET precipitation and temperature fields to the basin scale through bias correction, using ground-based observations from DGA meteorological stations distributed across the Salar del Huasco basin (see Figs. B1 and B2). EEFLUX evaporation estimates were temporally disaggregated to daily resolution and further corrected through a machine learning approach constrained by local meteorological inputs (Figs. B3 and B4)."*

Lines 413-425: *"Second, our estimates are subject to uncertainties associated with satellite-derived input data and our interpolation approach (Appendix B). Although local bias corrections were applied using available observations, residual errors persist and inevitably propagate into the water balance analysis. At the station level, precipitation remains systematically overestimated, with percent bias values ranging from -0.3% to +26%, particularly at lower-elevation sites (Table B1, Fig. B2). When aggregated at the basin scale, the median signed bias is -10%. Based on this analysis and previous assessments of CR2MET performance, we adopt ±10% as a representative uncertainty bound for precipitation. For evaporation, while the temporal behavior of EEFLUX was validated against ground-based pan observations at a representative site in the salt flat nucleus (Fig. B5), the spatial coverage of ground data was insufficient to directly assess uncertainty across the basin. We therefore adopt a conservative uncertainty bound of 20%, consistent with validation studies of EEFLUX/METRIC in arid and high-elevation regions reporting typical errors of ∼5-20% (Nisa et al., 2021, Wasti, 2020, Lima et al., 2020, Madugundu et al., 2017). Even under the most favorable case, precipitation increased by 10% and evaporation decreased by 20%, the imbalance persists. Moreover, given that precipitation is more likely overestimated, the reported deficit is likely conservative, and the true imbalance may be larger. Thus, we do not attribute the imbalance primarily to data uncertainty."*

Lines 463-465: *"To this end, further validation of satellite-derived inputs against local observations across the basin, particularly for evaporation and air temperature, is strongly recommended to enhance the robustness of the water balance assessment."*

Lines 499-500: *"Finally, validating satellite-derived inputs against additional local observations is essential to strengthen the reliability of the water balance assessment."*

Lines 540-542: *"Since we required an accurate description of precipitation and air temperature, we proposed a machine learning based approach for bias correction using ground-based observations from the Chilean National Water Division (Dirección General de Aguas, DGA)."*

Lines 559-560: *"Tables B1 and B2 summarize the metadata of the precipitation and temperature stations used in the bias correction, together with performance metrics that compare ground-based observations against the corrected CR2MET datasets".*

Lines 610-619: *"We assessed EEFLUX performance by comparing its actual evaporation estimates with pan-derived potential evaporation, adjusted using a 0.7 coefficient as recommended by DGA (2009). These observations were collected at a representative site within the salt flat nucleus sub-basin (GP Consultores, 2008), the area with the highest evaporative fluxes in the study site. As expected, observed potential*

*evaporation exceeds EEFLUX-derived actual evaporation by 18% (Fig. B5), consistent with the differences between potential and actual evaporation. While this comparison is not suited for validating absolute magnitudes, it provides useful insight into the temporal behavior of EEFLUX, which reproduces both seasonal and daily variability captured by the pan observations well (Fig. B5). However, due to the limited spatial coverage and the nature of the observations, we consider a conservative ±20% uncertainty bound based on published studies (e.g., Nisa et al., 2021; Wasti, 2020; Lima et al., 2020; Madugundu et al., 2017) when propagating evaporation errors into the water balance analysis."*

Rainfall-runoff model

*Reviewer 2.* Lines 131-132: The authors state that the Uribe et al. (2015) rainfall-runoff model is driven solely by precipitation, and in the model setup, evaporation occurs only in response to rainfall events. Please clarify the rationale for selecting this model despite this limitation, and discuss how this assumption may affect the results and the interpretation of the water balance.

*Reviewer 1.* The discussion of the seasonal dynamics of groundwater recharge depends on modelling results, but the model was calibrated using only surface flows. It seems that the groundwater dynamics are largely unconstrained. Some validation with local groundwater data would be invaluable. If not, at least there should be appropriate caveats.

*Reviewer 1.* Fig 3 results. The model that was used to simulate groundwater recharge was calibrated on observed river flows. So this provides only very limited information to define the dynamics of groundwater recharge fluxes. Were there no groundwater observations available to calibrate/validate this important component?

*Reviewer 2.* The Discussion section would benefit from a more detailed consideration of the limitations associated with (…) the assumptions of the applied model. (…) it is important to discuss the potential uncertainties arising from (…) model parameterization, and structural assumptions.

We used the Uribe et al. (2015) rainfall-runoff model because it was originally developed for the Salar de Huasco and is based on hydrological response units (HRUs) already defined for this basin and widely used in Altiplano studies (e.g., Acosta, 2004; Blin et al., 2022; Yañez-Morroni et al., 2024a,b). This ensures consistency with earlier work and takes advantage of a model structure already adapted to the physical, geological, and climatic conditions of the catchment. While the original model formulation represents evaporation mainly as an event-driven process tied to rainfall, we adapted the setup to better capture off-rainy season evaporation by introducing an evaporation component that operates independently of rainfall inputs. This allows the model to account for evaporation during the long dry season, which is particularly important in high-altitude basins such as the Salar de Huasco. In our study, the model is therefore used primarily to represent the partitioning and timing of surface runoff and groundwater recharge, while basin-scale closure is derived independently from bias corrected satellite-based evaporation and precipitation, considering that no changes in groundwater levels have been observed in the basin (see Blin et al., 2022). However, we now acknowledge in the revised manuscript that the modified model may misrepresent water losses, particularly during the dry season, due to uncertainties in satellite-derived evaporation data, which could in turn impact the accuracy of groundwater recharge estimates.

We also agree that the recharge component is less constrained as calibration relies solely on surface flow records. Local groundwater observations do exist, but they are sparse and spatially discontinuous, making them unsuitable for direct quantitative calibration or validation of groundwater recharge estimates. For this

reason, we opted not to compare model outputs with these heterogeneous datasets. However, although not validated here against groundwater levels, the same model has been applied in the Salar del Huasco and other basins, where its recharge estimates were incorporated into groundwater flow models that performed well (Blin et al., 2022; Blin & Suárez, 2023). For instance, the Uribe et al. (2015) model was used by Blin et al. (2022) to estimate aquifer recharge in the basin, which was then further employed to drive the groundwater model that represented well the water levels observed throughout the basin (see simulated and observed well levels in Blin et al., 2022). Therefore, although this model is calibrated with river discharge, it does provide a plausible spatiotemporal distribution of recharge that is consistent with observations from groundwater wells in the Salar de Huasco aquifer. This gives confidence that our recharge outputs are reasonable as first-order estimates.

Finally, we also acknowledge that our rainfall-runoff modeling framework is subject to sources of uncertainty and key model assumptions. We now discuss their potential impact on model results:

1. Satellite-derived input limitations: Despite bias corrections based on ground observations, residual errors remain in satellite-derived precipitation, temperature, and evaporation fields, particularly in sparsely monitored areas. These uncertainties propagate through the model and can lead to misrepresentation of water inputs and outputs, thereby influencing the estimated recharge and balance outcomes.

2. Dry-season evaporation parameterization: Evaporation during the dry season is not dynamically simulated but instead prescribed using satellite-derived input. This may lead to under- or overestimation of seasonal water losses, which can directly affect recharge estimates, especially during the dry season.

3. Omission of cryospheric and freeze-thaw processes: The model does not account for snow accumulation, snowmelt, or soil freeze-thaw processes which, although not dominant, have been observed in the Salar del Huasco basin. Excluding these processes may lead to inaccurate timing and magnitude of infiltration and recharge, especially during transitional seasons (e.g., spring melt), potentially biasing recharge estimates.

4. Streamflow data limitations: The discharge data used for model calibration are subject to observational uncertainty, especially during low or ephemeral flow conditions. These limitations can reduce the reliability of the calibration and impact the model's ability to correctly represent water partitioning processes, including infiltration and runoff.

5. Model parameter uncertainty: The model includes 15 calibrated parameters, increasing the risk of equifinality, where multiple parameter sets yield similar performance but differ in internal process representation. This compromises the uniqueness and robustness of the estimated recharge. Following recent recommendations (e.g., Yáñez-Morroni et al., 2024a,b), we now suggest that future work simplify the model by reducing the number of parameters and using less complex formulations, to improve the clarity and reliability of parameter estimates.

We would like to emphasize that, despite the limitations in the rainfall-runoff model, it offers a spatially distributed and process-oriented framework for analyzing groundwater recharge, and it enables exploration of the dominant hydrological controls in the Salar del Huasco basin. Nevertheless, we now emphasize that

the results should be interpreted as indicative, and future refinement will benefit from improved data coverage and model structure.

To strengthen the revised manuscript:

1. We now clarify the role of the model: We explain the rationale for selecting this model and explicitly state that the model provides inferred recharge fluxes based on surface-water calibration, rather than direct simulations of groundwater levels. We also emphasize that the model is primarily used to represent the partitioning and timing of surface runoff and recharge.

2. Note previous successful applications: We now highlight in the manuscript that the same model structure has been successfully used in the Salar del Huasco and other basins, where its recharge estimates were incorporated into groundwater models (e.g., MODFLOW in Blin et al., 2022), where observed water levels are well represented.

3. Add current limitations and future directions: We now discuss the implications of parameterization and assumptions in the rainfall-runoff model, as well as input data uncertainty. We also reinforce the ideas that (i) recharge estimates should be viewed as first-order approximations; and (ii) future work should integrate groundwater-level or tracer data where available to validate recharge dynamics.

We have modified the revised manuscript as follows:

Lines 106-112: *"Thus, the model provides inferred recharge fluxes based on surface-water calibration (Appendix A). This model has been previously applied in the Salar del Huasco and other Altiplano basins* (e.g., Yañez-Morroni et al., 2024a,b, Blin et al., 2022; Acosta, 2004)*, where its recharge outputs were successfully incorporated into groundwater flow models, demonstrating its credibility in groundwater applications. For instance, the Uribe et al. (2015) model was applied by Blin et al. (2022) to estimate aquifer recharge in the Salar de Huasco, providing a spatiotemporal distribution of recharge consistent with groundwater-well observations. Here, we primarily use it to represent the timing and partitioning of surface runoff and recharge."*

Lines 294-305: *"Despite these findings, it is important to acknowledge that our groundwater recharge estimates are subject to uncertainties related to both model structure and input data. The modified rainfall–runoff model offers a first-order approximation and relies on satellite-derived inputs that, despite local bias correction, retain spatial and temporal uncertainties, particularly in poorly monitored areas. These residual errors can propagate through the model and affect recharge estimates. In addition, dry-season evaporation is prescribed using satellite estimates rather than dynamically simulated, which may misrepresent seasonal water losses and bias recharge estimates under dry conditions. The model also excludes snow accumulation, melt, and freeze-thaw dynamics, which, while not dominant, may influence infiltration timing in high-altitude areas. Furthermore, the 15-parameter calibration increases the risk of equifinality, potentially limiting the uniqueness of the recharge estimates. Lastly, recharge fluxes should be interpreted as model-inferred outputs constrained by surface flows rather than direct simulations of groundwater levels. Despite these limitations, the model enables spatially distributed and process-oriented analysis of groundwater recharge, and provides a valuable baseline for understanding the hydrological dynamics in the Salar del Huasco basin under the current climatic and data constraints."*

Lines 303-305: *"Lastly, recharge fluxes should be interpreted as model-inferred outputs constrained by surface flows rather than direct simulations of groundwater levels."*

Lines 493-495: *"Also, incorporating groundwater-level or tracer data, where available, is strongly recommended to provide independent validation of the recharge estimates. Furthermore, refining the model structure to reduce parameterization, as demonstrated in the Silala basin (Yañez-Morroni et al., 2024a,b), would help minimize parameter uncertainty and enhance the robustness of groundwater applications."*

Lines 511-527: *"Limitations must be considered when interpreting the results of the modified rainfall-runoff model. First, evaporation during the dry season is not dynamically simulated but rather prescribed based on satellite-derived estimates. This simplification can lead to under- or overestimation of seasonal water losses and, consequently, affect recharge estimates, particularly during low-flow periods. Second, the model relies on satellite-derived precipitation, temperature, and evaporation inputs that, despite being bias-corrected, retain residual uncertainties, especially in regions with sparse ground observations. These uncertainties propagate through the model and may influence the accuracy of water balance components. Third, key high-altitude hydrological processes such as snow accumulation, melt runoff, and freeze-thaw soil dynamics are not represented. While not dominant, these processes have been observed in the Salar del Huasco and may influence the timing and magnitude of infiltration and recharge. Additionally, calibration is based on observed streamflows that are subject to measurement uncertainty, particularly during periods of low or ephemeral flow, which can affect model performance assessments. Lastly, the use of 15 calibrated parameters increases the risk of equifinality, where different parameter combinations yield similar model performance but diverge in internal process representation. This limits confidence in the uniqueness and robustness of the simulated recharge. Following recent recommendations (e.g., Yáñez-Morroni et al., 2024a,b), future work should aim to reduce model complexity and improve parameter identifiability through more parsimonious formulations. Despite these limitations, the model provides a spatially distributed, process-oriented framework that enables exploration of the dominant hydrological drivers in the Salar del Huasco basin. The results should be interpreted as indicative, with future refinements benefiting from improved observational data and enhanced model structure."*

Other minor comments Reviewer 1

*Reviewer 1.* Line 13: replace 'insinuate' by 'imply'

We have updated the revised manuscript in line 13.

*Reviewer 1.* Line 37: composed of

We have updated the revised manuscript in line 38.

*Reviewer 1.* Line 91 of the Uribe et al…

Line 112 the Uribe….

Line 131 the Uribe…

Line 156 the Uribe… please correct throughout – including Fig 3 caption

We have added "the" before "Uribe et al." throughout the manuscript, including Figure 3 caption.

*Reviewer 1.* Line 94: into

We have updated the revised manuscript in line 98.

*Reviewer 1.* Section 2.2 para 2 specify the model and forcing data time steps. I assume daily??

We have clarified that model and forcing data time steps are daily.

Lines 94-95: *"The rainfall-runoff model estimates key hydrological processes at a daily timescale by partitioning precipitation (…)".*

Lines 119-120: *"We first modified Uribe et al. (2015) model by integrating daily precipitation, air temperature and evaporation gridded data instead of using the original topographic-gradients approach".*

*Reviewer 1.* Line 116: data … are used…

We have updated the revised manuscript in line 123.

*Reviewer 1.* Line 118: please clarify what is meant by evaporation here. It isn't obvious until line 137.

We have clarified the definition in lines 126-127: *"In this study, evaporation refers specifically to satellite-retrieved actual evaporation from EEFLUX/METRIC, rather than potential evaporation or model-estimated fluxes."*

Other minor comments Reviewer 2

This manuscript presents an investigation of the spatiotemporal variability of groundwater recharge and upwelling in the Salar del Huasco basin, focusing on its interactions with precipitation, evaporation, and overall water balance dynamics in this arid endorheic system located in the Chilean Altiplano. The study addresses an important and timely topic with relevance to hydrological processes in data-scarce, high-altitude regions and provides valuable insights into the interactions among precipitation, evaporation, and groundwater recharge under conditions of extreme water scarcity. The manuscript is well structured, and the results are presented clearly. The work has the potential to make a contribution to the understanding of water balance processes in arid endorheic systems. However, certain aspects would benefit from clarification and further elaboration.

Some specific comments are provided below for consideration:

*Reviewer 2.* Lines 9-10: The message of this sentence is not entirely clear and would benefit from rephrasing to improve coherence and accuracy. Groundwater itself cannot "lead to recharge," since recharge is the process that feeds or replenishes groundwater storage. The intended meaning seems to be that groundwater storage decreases, or that recharge reaches a minimum following the dry period.

We agree and have rephrased the sentence for accuracy. The intended meaning is that following the dry period, groundwater sustains high evaporation rates, while recharge declines to a minimum.

Lines 9-11: *"Our findings highlight that when summer rainfall ceases, groundwater becomes the main water source supporting high evaporation rates, while recharge reaches a minimum by the end of autumn that persists until the end of the year."*

*Reviewer 2.* Line 11: The term *"competition"* is not ideal in this hydrological context. Please replace *"trade-off" or similar words* to describe the contrasting relationship between groundwater recharge and evaporation.

We agree and now use the term "trade-off" throughout the manuscript, which better reflects the contrasting relationship between recharge and evaporation.

Line 11: *"These results suggest a trade-off between groundwater recharge and evaporation for available water during the dry season."*

Line 67: *"We also further modify the model to study groundwater evaporation - recharge trade-off for moisture"*

Line 148: *"… improve estimates of groundwater recharge by accounting for the trade-off between recharge and evaporation for available moisture"*

*Reviewer 2.* lines 84-87 It would be helpful to express the total annual discharge in units of mm yr$^{-1}$, in addition to the volumetric flow rates (m$^3$ s$^{-1}$), to facilitate direct comparison with precipitation and evaporation values reported elsewhere in mm yr$^{-1}$ in the manuscript.

While we report spring flows in m$^3$s$^{-1}$, conversion to mm yr$^{-1}$ requires knowledge of the contributing recharge area, which is not available for these springs. To at least provide an order of magnitude, we used the salt-flat nucleus sub-basin area (48.22 km$^2$) as a reference. Under this assumption, the total spring discharge of ∼0.2 m$^3$ s$^{-1}$ corresponds to roughly 130 mm yr$^{-1}$.

We now report this order-of-magnitude estimate in the manuscript, while we continue to present the observed values in m$^3$s$^{-1}$ that can be directly compared to precipitation and evaporation fluxes also expressed in m$^3$/s in Figs. 6 and 7.

Lines 88-89: *"In total, they contribute to the salt-flat water balance with ∼0.2 m$^3$ s$^{-1}$, equivalent to an order of magnitude of ∼10$^2$ mm yr$^{-1}$."*

*Reviewer 2.* Figure 3: It is unclear how "monthly mean variability" can be represented in *mm yr$^{-1}$*, since those units denote annual totals or rates. It appears that Figure 3 presents monthly mean values on the x-axis; therefore, the corresponding y-axis units should likely be expressed in *mm* (or *mm month$^{-1}$*), rather than *mm yr$^{-1}$*. Please clarify the unit definition. If the intention is to show monthly distributions, the values should be expressed in *mm month$^{-1}$* (or simply *mm*), or the caption should explicitly state that the data have been annualized (e.g., monthly averages scaled to their annual equivalents). Please clarify the units and calculation method to ensure the figure is interpreted correctly. Otherwise, the comparison between rainfall and evaporation may be misleading, as expressing monthly values in *mm yr$^{-1}$* artificially inflates their magnitude by a factor of 12.

*Reviewer 2.* The results described between lines 207–225 also seem to be based on monthly values presented in Figure 3. If this is the case, the current labeling in *mm yr$^{-1}$* is inconsistent and may cause confusion. Conversely, if the data were converted to annual equivalents, it would be preferable to present Figure 3 using an annual scale rather than monthly intervals, to maintain conceptual consistency.

*Reviewer 2.* A similar applies to Figure 4, particularly the right panels (a–c). The values are presented as a function of months, suggesting that the data represent *monthly means*. If the intention is to illustrate the mean variation across available months (e.g., showing typical January, February, etc., values), then the units should correspond to the monthly time step. Alternatively, if the values have been annualized, please clarify this in the caption and consider adjusting the x-axis to represent annual rather than monthly time intervals for conceptual consistency.

The reviewer is correct that Figures 3 and 4, and also Figure 5, show monthly means on the x-axis. All water balance components in the manuscript (precipitation, evaporation, and recharge) are presented as monthly averages but expressed in annual-equivalent units (mm yr⁻¹). This rescaling was applied to keep a consistent vertical scale across fluxes and to avoid very small values in mm month⁻¹, particularly for recharge, which is our main focus. Thus, the x-axis remains monthly to highlight seasonal variability, while the y-axis in mm yr⁻¹ ensures comparability across figures. The captions now make this explicit:

For Fig. 3: *"Values are presented for the 1980 - 2019 period and represent monthly means, scaled to annual equivalents (mm year-1) for consistency across all water balance components."*

For Figs. 4 and 5: *"Values represent monthly means, scaled to annual equivalents (mm year-1) for consistency across all water balance components."*

We have also clarified this in the text:

Line 206-207*: "Groundwater recharge is displayed as monthly averages scaled to their annual equivalents in mm year⁻¹, along with precipitation and evaporation observations from the satellite-derived datasets."*

Line 242-243*: "Values are scaled to their annual equivalents in mm year⁻¹".*

*Reviewer 2.* Line 196 -206: In this section, the results are presented without a corresponding reference to any figure or table. Please specify which figure (or sub-figure) illustrates these findings so that readers can easily locate and interpret the results. Clear cross-referencing between text and figures would greatly improve readability and traceability of the analysis for the Results and Discussion section

We have added explicit cross-references to Fig. 3 where the results are discussed, and we have also carefully reviewed the manuscript to ensure that all figure references are consistent.

*Reviewer 2.* Line 244: The text refers to "S1-type HRUs with an annual mean of ∼600 mm yr⁻¹ (see Fig. 2)," but Figure 2 does not display any numerical values or spatial distribution of annual means; it only shows the HRU classification. Please clarify whether these mean values are derived from another figure, dataset, or analysis step, and adjust the figure reference accordingly.

We thank the reviewer for this correction. We corrected this reference to Fig. 4b in line 260.

**References**

Acosta, O. (2004). Impacto de las extracciones de agua subterránea en el salar del huasco. Master's thesis, Tesis de Máster, Universidad Politécnica de Cataluña. Barcelona, España.

Blin, N., Hausner, M., Leray, S., Lowry, C., Suárez, F. (2022). Potential impacts of climate change on an aquifer in the arid altiplano, northern chile: The case of the protected wetlands of the salar del huasco basin. Journal of Hydrology: Regional Studies 39, 100996. https://doi.org/10.1016/j.ejrh.2022.100996

Blin, N., & Suárez, F.. (2023). Evaluating the contribution of satellite-derived evapotranspiration in the calibration of numerical groundwater models in remote zones using the EEFlux tool, Science of The Total Environment, Volume 858, Part 1, 159764, ISSN 0048-9697, https://doi.org/10.1016/j.scitotenv.2022.159764.

Collahuasi - GP Consultores. (2008). Información sobre evaporación provista por Compañía Minera Doña Inés de Collahuasi. Informes inéditos.

DGA. (2009). Sistema Piloto I Región: Salar del Huasco. In Levantamieno hidrogeológico para el desarrollo de nuevas fuentes de agua en áreas prioritarias de la zona norte de Chile, Regiones XV, I, II y III.

Madugundu, R., Al-Gaadi, K. A., Tola, E., Hassaballa, A. A., and Patil, V. C. (2017). Performance of the METRIC model in estimating evapotranspiration fluxes over an irrigated field in Saudi Arabia using Landsat-8 images, Hydrol. Earth Syst. Sci., 21, 6135–6151, https://doi.org/10.5194/hess-21-6135-2017.

Nisa, Z., Khan, M. S., Govind, A., Marchetti, M., Lasserre, B., Magliulo, E., & Manco, A. (2021). Evaluation of SEBS, METRIC-EEFlux, and QWaterModel Actual Evapotranspiration for a Mediterranean Cropping System in Southern Italy. *Agronomy*, *11*(2), 345. https://doi.org/10.3390/agronomy11020345

Lima, João & Sánchez-Tomás, Juan & G. Piqueras, José & Sobrinho, José & C. Viana, Paula & S. Alves, Aline. (2020). Evapotranspiration of sorghum from the energy balance by METRIC and STSEB. Revista Brasileira de Engenharia Agrícola e Ambiental. 24. 24-30. 10.1590/1807-1929/agriambi.v24n1p24-30.

Wasti, Shailaja. (2020). Estimation of land surface Evapotranspiration in Nepal using Landsat based METRIC model. 10.48550/arXiv.2007.13922.

Yáñez-Morroni, G., Suárez, F., Muñoz, J.F., Lagos, M.S. (2024a). Hydrological modeling of the silala river basin. 1. model development and long-term groundwater recharge assessment. WIREs Water 11(1), 1690 https://doi.org/10.1002/wat2.1690

Yáñez-Morroni, G., Suárez, F., Muñoz, J.F., Lagos, M.S. (2024b). Hydrological modeling of the silala river basin. 2. validation of hydrological fluxes with contemporary data. WIREs Water 11(1), 1696 https://doi.org/10.1002/wat2.1696

---

## Author Response (AR2)

**Responses to Minor Revisions of Egusphere-2025-2984: "Questioning the Endorheic Paradigm: Water Balance dynamics in the Salar del Huasco basin, Chile" By Francisca Aguirre-Correa et al.**

We would like to once again thank the Reviewers for their valuable feedback throughout the review process and for recommending our manuscript for publication in HESS. We also extend our sincere thanks to Dr. Gabriel Rau for efficiently handling the review process and for his constructive revisions.

All minor comments from Dr. Gabriel Rau and Prof. Howard Wheater have been addressed in the attached revised manuscript. Our responses to the comments are provided in blue font, indicating how and where the revised manuscript has been modified. In addition, we include a second version of the manuscript in which all changes are highlighted for clarity.

Editor Dr. Gabriel Rau

*Editor.* The Figure 3: It would be more intuitive to state the y-axes tick numbers on the right side as full numbers (not scientific notation). Also, a legend would help to interpret the blue and red shades shown in the plot (in addition to the caption explaining these).

We have updated Figure 3 accordingly by using full-number y-axis tick labels and adding a legend to clarify the blue and red shading as La Niña and El Niño years, respectively.

*Editor.* Figures 4 and 5: Labelling of tick marks of the colour bars (left side) and subplots (right side) should also be full numbering (e.g., 1,200 instead of 1.2 x 10^3) to improve intuitive comparison of the numbers.

We have updated Figures 4 and 5 so that all tick labels are shown as full numbers rather than in scientific notation.

*Editor.* In the Conclusions, please could you place a line break before "Beyond these dynamics ..." to separate the two paragraphs and their logic.

We have inserted the requested line break in line 491.

*Editor.* Data availability: Stating "Data used in this research is publicly accessible" is not sufficient. Please organise and archive the dataset you compiled in a suitable repository (e.g., figshare, Zenodo, etc.) with description and permanent DOI, and cite this in the data availability section.

We have archived the compiled dataset in a public repository (DOI: https://doi.org/10.17605/OSF.IO/6BG2W) and updated the Data Availability section to include the repository link and citation.

Lines 641-643: *"Data availability. The downscaled datasets generated for each Hydrological Response Unit (HRU) defined in this study are openly available at: https://doi.org/10.17605/OSF.IO/6BG2W. The original datasets used as input for the downscaling are publicly available from their respective repositories."*

Reviewer 1 Prof. Howard Wheater

*Reviewer 1.* Line 52 delete comma

We have updated the revised manuscript in line 52.

*Reviewer 1.* Line 106 suggest insert 'while of lack of reliable groundwater data precludes explicit calibration of groundwater recharge, we note that …this model has previously….'

We have included the suggested sentence in lines 107-110 of the revised manuscript.

Lines 107-110: *"While the lack of reliable groundwater data precludes explicit calibration of groundwater recharge, we note that this model has been previously applied in the Salar del Huasco and other Altiplano basins (e.g., Yáñez-Morroni et al. (2024a, b); Blin et al. (2022); Acosta (2004)). In these studies, recharge outputs from the rainfall-runoff model were successfully incorporated into groundwater flow models, demonstrating its credibility in groundwater applications."*

*Reviewer 1.* Line 127 the satellite derived estimates are in fact model based! Minor rephrase needed e.g. 'or other model-based estimates'

We have rephrased the sentence in the revised manuscript.

Line 128: *"… rather than potential evaporation or other model-estimated fluxes."*

*Reviewer 1.* L152 reservoir's

We have updated the revised manuscript in line 152.

*Reviewer 1.* L154 not simulated explicitly

We have updated the revised manuscript in lines 154-155.

*Reviewer 1.* L236 what is meant by overall hydrological response? Please clarify

The basin hydrological response was already defined in the manuscript as the sum of the individual HRU responses across sub-basins (lines 170-171 in the original manuscript, lines 171-172 in the revised manuscript). To avoid ambiguity, we have removed the term "overall" in line 240.

*Reviewer 1.* L344 replace 'remind' with 'remember'

We have updated the revised manuscript in line 341.

*Reviewer 1.* L395 delete first comma

We have updated the revised manuscript in line 393.

*Reviewer 1.* L412 associated with

We have updated the revised manuscript in line 410.

*Reviewer 1.* L416 replace 'such' with 'the large'

We have updated the revised manuscript in line 414.

*Reviewer 1.* L471 although this is recommended in the conclusions, I suggest including here a recommendation for gw observations to constrain the hydrological model and groundwater recharge estimates. It seems odd not to include it in this discussion.

We have added a recommendation at the end of the Discussion section highlighting the need for groundwater observations to better constrain the hydrological model and groundwater recharge estimates.

Lines 465-470: *"To this end, groundwater observations are essential to better constrain the hydrological model and groundwater recharge estimates. Further validation of satellite-derived inputs against local observations across the basin, particularly for evaporation and air temperature, is also strongly recommended to enhance the robustness of the water balance assessment. In addition, a more integrated and robust methodological approach is recommended to quantify subsurface storage changes and to estimate how large the groundwater catchment would need to be to explain the observed water balance of the basin."*